

# Solving optimization problems simultaneously: the variants of the traveling salesman problem with time windows using multifactorial evolutionary algorithm

Ha-Bang Ban and Dang-Hai Pham

Computer Science Department, School of Information and Communication Technology, Hanoi University of Science and Technology, Hanoi, Vietnam

## ABSTRACT

We studied two problems called the Traveling Repairman Problem (TRPTW) and Traveling Salesman Problem (TSPTW) with time windows. The TRPTW wants to minimize the sum of travel durations between a depot and customer locations, while the TSPTW aims to minimize the total time to visit all customers. In these two problems, the deliveries are made during a specific time window given by the customers. The difference between the TRPTW and TSPTW is that the TRPTW takes a customer-oriented view, whereas the TSPTW is server-oriented. Existing algorithms have been developed for solving two problems independently in the literature. However, the literature does not have an algorithm that simultaneously solves two problems. Multifactorial Evolutionary Algorithm (MFEA) is a variant of the Evolutionary Algorithm (EA), aiming to solve multiple factorial tasks simultaneously. The main advantage of the approach is to allow transferrable knowledge between tasks. Therefore, it can improve the solution quality for multitasks. This article presents an efficient algorithm that combines the MFEA framework and Randomized Variable Neighborhood Search (RVNS) to solve two problems simultaneously. The proposed algorithm has transferrable knowledge between tasks from the MFEA and the ability to exploit good solution space from RVNS. The proposed algorithm is compared directly to the state-of-the-art MFEA on numerous datasets. Experimental results show the proposed algorithm outperforms the state-of-the-art MFEA in many cases. In addition, it finds several new best-known solutions.

# INTRODUCTION

## The TSPTW and TRPTW literature

The Traveling Salesman Problem with Time Windows (TSPTW) (*Gendreau et al., 1998*; *Silva & Urrutia, 2010*; *Ohlmann & Thomas, 2007*), and Traveling Repairman Problem with Time Windows (TRPTW) (*Ban & Nghia, 2017*; *Ban, 2021*; *Heilporna, Cordeaua & Laporte, 2010*) are combinatorial optimization problems that have many practical situations. The TRPTW wants to minimize the sum of travel durations between a depot

Corresponding author
Ha-Bang Ban,
bangbh@soict.hust.edu.vn

and customer locations, while the TSPTW aims to minimize the total time to visit all customers. In the two problems, the deliveries are made during a specific time window given by the customers. Due to time window constraints, the TSPTW and TRPTW are much harder than the traditional Traveling Salesman Problem (TSP) and Traveling Repairman Problem (TRP).

The Travelling Salesman Problem with Time Windows (TSPTW) is a popular NP-hard combinatorial optimization problem studied much in the literature (*Dumas, Desrosiers & Gélinas, 1995*). The algorithms include exact and metaheuristic approaches. *Langevin et al. (1993)* introduced a two-commodity flow formulation to solve the problem. *Dumas, Desrosiers & Gélinas (1995)* then used a dynamic programming approach. Similarly, *Gendreau et al. (1998)* brought constraint programming and optimization techniques together. Most recently, *Dash et al. (2012)* propose a method using an IP model based on the discretization of time. The results are extremely good: several benchmark instances are solved first. *Gendreau et al. (1998)* then proposed an insertion heuristic that generated the solution in the first step and improved it in a post-phase using removal and reinsertion of vertices. *Ohlmann & Thomas (2007)* developed simulated annealing relaxing the time windows constraints by integrating a variable penalty method within a stochastic search procedure. In this work, they developed a two-phase heuristic. In the first phase, a feasible solution was created by using a Variable Neighborhood Search (VNS) (*Mladenovic & Hansen, 1997*). In the post phase, this solution was improved by using a General VNS. Generally speaking, the results from this approach are very promising.

In the Travelling Repairman Problem with Time Windows (TRPTW), there is an exact algorithm and three metaheuristics algorithms in the literature: (1) *Tsitsiklis (1992)* proposed a polynomial algorithm when several customers are bounded; (2) *Heilporna, Cordeaua & Laporte (2010)* then developed an exact algorithm and heuristic algorithm to solve the problem; (3) *Ban & Nghia (2017)* and *Ban (2021)* proposed two metaheuristic algorithms based on Variable Neighborhood Search (VNS) scheme. Their experimental results showed the efficiency of the metaheuristic approach.

These above algorithms are the state-of-the-art metaheuristics in current. However, they are designed to solve each problem independently. That means they cannot solve both two problems well simultaneously. In *Salehipour et al. (2011)* also showed that a good algorithm for this problem might not be good for the other problem. Therefore, developing an algorithm solving both problems well simultaneously is our aim in this work.

## The MFEA literature

To date, the MFEA framework (*Dash et al., 2012*; *Gupta, Ong & Feng, 2016*; *Xu et al., 2021*) has been introduced in the literature. Using a unified search space for multiple tasks can exploit good transferrable knowledge between optimization tasks. In addition, although solving many tasks simultaneously, the flow of the MFEA framework is sequential. Therefore, the MFEA is suitable in a system with limited computation.

Several close variants of the MFEA (*Ban & Pham, 2022*; *Osaba et al., 2020*; *Yuan et al., 2016*) are developed to solve permutation-based optimization problems such as the TSP and TRP. Therefore, they are direct references to our research. *Yuan et al. (2016)* firstly

developed evolutionary multitasking in permutation-based optimization problems. They tested it on several popular combinatorial problems. The experiment results indicated the good scalability of evolutionary multitasking in many-task environments. *Osaba et al. (2020)* then proposed a dMFEA-II framework to exploit the complementarities among several tasks, often achieved *via* genetic information transfer. Their dMFEA-II controls the knowledge transfer by adjusting the crossover probability value. The technique allows good knowledge to transfer between tasks. Although the results are promising, the drawback of the above two algorithms (*Osaba et al., 2020*; *Yuan et al., 2016*) is that there is a lack of a mechanism to exploit the good solution space explored by MFEA. Therefore, these algorithms cannot effectively balance exploration and exploitation (we visualize their issue in "Comparison with the previous MFEA algorithms" in more detail). Recently, *Ban & Pham (2022)* have successfully applied the MFEA with RVNS to solve two problems such as TSP and TRP. Their algorithms maintain the exploration and exploitation better than the others in *Osaba et al. (2020)* and *Yuan et al. (2016)*. Its performance encourages us to use this combination to solve the TSPTW and TRPTW. This article considers these works as a baseline for our research.

## Our contributions

This article introduces the first algorithm combining the MFEA framework and RVNS to solve two tasks simultaneously. The combination is to have positive transferrable knowledge between tasks from the MFEA and the ability to exploit good solution spaces from RVNS. The major contributions of this work are as follows:

- We propose a new selection operator that balances skill-factor and population diversity. The skill-factor effectively transfers elite genes between tasks, while diversity in the population is important when it meets a bottleneck against the information transfer.
- Multiple crossover schemes are applied in the proposed MFEA. They help the algorithm have good enough diversity. In addition, two types of crossover (intra- and inter-) are used. It opens up the chance for knowledge transfer through crossover-based exchange between tasks.
- The combination between the MFEA and the RVNS is to have good transferrable knowledge between tasks from the MFEA and the ability to exploit good solution spaces from the RVNS. However, focusing only on reducing cost function maybe lead the search to infeasible solution spaces like the algorithms (*Ban & Pham, 2022*; *Osaba et al., 2020*; *Yuan et al., 2016*). Therefore, the repair method is incorporated into the proposed algorithm to balance finding feasible solution spaces and reducing cost function.
- Numerical experiments show that the proposed algorithm reaches nearly optimal solutions simultaneously in a short time for two problems. Moreover, it obtains better solutions than the previous MFEA algorithms in many cases.

The rest of this article is organized as follows. Sections 1 and 2 present the literature and preliminary, respectively. Section 3 describes the proposed algorithm. Computational evaluations are reported in Section 4. Section 5 presents the conclusions and future work.

# THE FORMULATION AND METHODOLGY

## The formulation

We consider an example that describes the difference between two problems in a specific instance. If we use the optimal solution of n40w160.002 instance for the TSPTW (https://lopez-ibanez.eu/tsptw-instances), the objective function cost (using the function cost of the TRPTW) of this solution for the TRPTW is 7,519, while the known-best cost for this instance for the TRPTW is 6,351 (the known-best cost is found by our algorithm). Thus, the difference between the two objective function costs is 15.5%. It implies that a good metaheuristic algorithm for the TSPTW does not produce a good solution for the TRPTW and vice versa. The above algorithms are the best algorithms for two problems. However, they only solve each problem independently but cannot simultaneously produce good solutions for two problems.

We have a complete graph $K_n = (V, E)$, where $V = v_1, v_2, \ldots, v_n$ is a set of vertices showing the starting vertex and customer locations, and $E$ the set of edges connecting the customer locations. Suppose that, in a tour $T = (v_1 = s, v_2 \ldots, v_n)$, each edge $(v_i, v_j) \in E$ connecting the two vertices $v_i$ and $v_j$ there exists a cost $c(v_i, v_j)$. This cost represents the travel time between vertex $v_i$ and $v_j$. Each vertex $v_i \in V$ has a time window $[e_i, l_i]$ indicating when starting service time at vertex $v_i$. This implies that a vertex $v_i$ may be reached before the start of $e_i$, but the service cannot start until $e_i$ and no later than $l_i$ of its time window. Moreover, to serve each customer, the salesman spends an amount of time. Let $D(v_i), S(v_i)$ be the time at which service begins and the service time at vertex $v_i$. It is calculated as follows: $D(v_i) = \max\{A(v_i), e_i\}$, where $A(v_k) = D(v_{i-1}) + S(v_{i-1}) + c(v_{i-1}, v_i)$ is the arrival time at vertex $v_i$ in the tour. A tour is feasible if and only if $A(v_i) \leq l_i$ for all vertices. The objective functions of the two problems are defined as follows:

- In the TSPTW, the salesman must return to $s$. Therefore, the cost of the tour $T$ is defined as: $\sum_{i=1}^{n} c(v_i, v_{i+1})$. Note that: $v_{n+1} \equiv s$
- In the TRPTW, we also define the travel duration of vertex $v_i$ as the difference between the beginning of service at vertex $v_i$ and the beginning of service at $s$: $t_i = D(v_i) - D(s)$. The cost of the tour $T$ is defined: $\sum_{i=2}^{n} t_i$.

Two problems consist of determining a tour, starting at the starting vertex $v_1$, minimizing the cost of the tour overall vertices while respecting time windows. First, note that: the man must start and end at vertex $v_1$.

## Our methodology

For NP-hard problems, we have three approaches to solve the problem, specifically, (1) exact algorithms, (2) approximation algorithms, and (3) heuristic (or metaheuristic) algorithms:

- The exact approaches find the optimal solution. However, they are exponential time algorithms in the worst case.
- An $\alpha$-approximation algorithm generates a solution that has a factor of $\alpha$ of the optimal solution.

- Heuristic (metaheuristic) algorithms perform well in practice and validate their performance through experiments. This approach is suitable for a problem with large sizes.

Previously, several metaheuristics have been proposed to solve the TSPTW (*Dumas, Desrosiers & Gélinas, 1995*; *Focacci, Lodi & Milano, 2002*; *Gendreau et al., 1998*; *Ohlmann & Thomas, 2007*; *Yuan et al., 2016*) and the TRPTW (*Ban & Nghia, 2017*; *Ban, 2021*; *Heilporna, Cordeaua & Laporte, 2010*; *Tsitsiklis, 1992*). However, they are developed to solve each problem independently and separately. Therefore, they cannot solve both two problems well simultaneously. When we run the two best algorithms for two tasks independently, there is no transferrable knowledge between tasks, and we cannot improve solution quality.

This article proposes an MFEA approach to solve two problems simultaneously. Our MFEA solves two tasks simultaneously: the first task is the TRPTW, and the second is the TSPTW. Experiment results indicate its efficiency: (1) for small instances, the proposed algorithm obtains the optimal solutions in both two problems; (2) for large ones, our solutions are close to the optimal ones, even much better than those of the previous MFEA approaches.

## The MFEA framework introduction

The overview of multifactorial optimization is introduced in *Dash et al. (2012)* and *Gupta, Ong & Feng (2016)*. Assume that $k$ optimization problems are needed to be performed simultaneously. Without loss of generality, tasks are assumed to be minimization problems. The $j$-th task, denoted $T_i$, has objective function $f_j$: $X_j \Rightarrow R$, in which $x_j$ is solution space. We need to be found $k$ solutions $\{x_1, x_2, …, x_{k-1}, x_k\} = \min\{f_1(x), f_2(x), …., f_{k-1}(x), f_k(x)\}$, where $x_j$ is a feasible solution in $X_j$. Each $f_j$ is considered an additional factor that impacts the evolution of a single population of individuals. Therefore, the problem also is called the $k-$ factorial problem. For the problem, a general method to compare individuals is important. Each individual $p_i(i \in \{1, 2, …, |P|\})$ in a population $P$ has a set of properties as follows: Factorial Cost, Factorial Rank, Scalar Fitness, and Skill Factor. These properties allow us to sort and select individuals in the population.

- Factorial Cost $c_j^i$ of the individual $p_i$ is its fitness value for task $T_j$ $(1 \leq j \leq k)$.
- Factorial rank $r_j^i$ of $p_i$ on the task $T_j$ is the index in the set of individuals sorted in ascending order in terms of $c_j^i$.
- Scalar-fitness $\phi_i$ of $p_i$ is given by its best factorial rank overall tasks as $\phi_i = \frac{1}{\min_{j \in 1,…,k} r_j^i}$.
- Skill-factor $\rho_i$ of $p_i$ is the one task, amongst all other tasks, on which the individual is most effective, *i.e.*, $\rho_i = argmin_j\{r_j^i\}$ where $j \in \{1, 2, …, k\}$.

The pseudo-code of the basic MFEA is described in Algorithm 1 (*Ban & Pham, 2022*): We first build the unified search space that encompasses all individual search spaces of different tasks to have a shared background on which the transfer of information can take place. We then initialize $SP$ individuals ($SP$ is the size of population) in the unified search space and then evaluate it by calculating the skill-factor of each individual. After the

**Algorithm 1** The basic MFEA.

1: Build Unified Search Space-USS;

2: Generate a population $p$;

3: Evaluate skill-factor and scalar-fitness in $P$;

4: **while** the stop criteria is not satisfied **do**

5:   $O \leftarrow \phi$;

6:   **while** $|O| \leq N$ **do**

7:     Sample two individuals $p_a$ and $p_b$ randomly from top half of parent population;

8:     **if** $\tau_a == \tau_b$ **then**

9:       $o_a, o_b \leftarrow$ Intra-task crossover on $p_a$ and $p_b$

10:       Assign offspring $o_a$ and $o_b$ skill factor $\tau_a$

11:     **else if** $rand < rmp$ **then**

12:       $o_a, o_b \leftarrow$ Intra-task crossover on $p_a$ and $p_b$

13:       Randomly assign offspring $o_a, o_b$ skill factor $\tau_a$ or $\tau_b$

14:     **else**

15:       $o_a \leftarrow$ mutation $(p_a)$;

16:       $o_b \leftarrow$ mutation $(p_b)$;

17:       $o_a, o_b$ have the same skill-factor as $p_a$ and $p_b$, respectively;

18:     $O = O \cup \{o_a, o_b\}$;

19:   $p = p \cup \{O\}$;

20:   Evaluate skill-factor and scalar-fitness in $P$;

21:   Elitism-Selection $(P)$;

initialization, the iteration begins to produce the offsprings and assign skill-factors to them. Selective evaluation guarantees that the skill-factor of each new offspring is selected randomly among those of the parents. The offspring and the parent are merged in a new population with $2 \times SP$ individuals. The evaluation for each individual is taken only on the assigned task (in the step, the best solution for each task is updated if it is found. This best solution for each task is the output). After evaluation, the individuals of the population receive new skill-factors. The Elitist strategy keeps the $SP$ solutions with the best skill-factors for the next generation.

Figure 1 (*Ban & Pham, 2022*) also shows the differences between the traditional EA and MFEA. The crossover and mutation operators in the MFEA are like the traditional EA. However, there are two different important aspects: (1) the parents' skill-factor and (2) random mating probability (*rmp*). Specifically, the child is created using crossover from parents with the same skill-factor. Otherwise, the child is generated by a crossover with a *rmp* value or by a mutation when parents own different skill-factors. A large *rmp* value generates more information exchanging between tasks. Also, in the traditional GA, the fitness of child is evaluated directly, while the skill-factor is assigned to it in the MFEA.

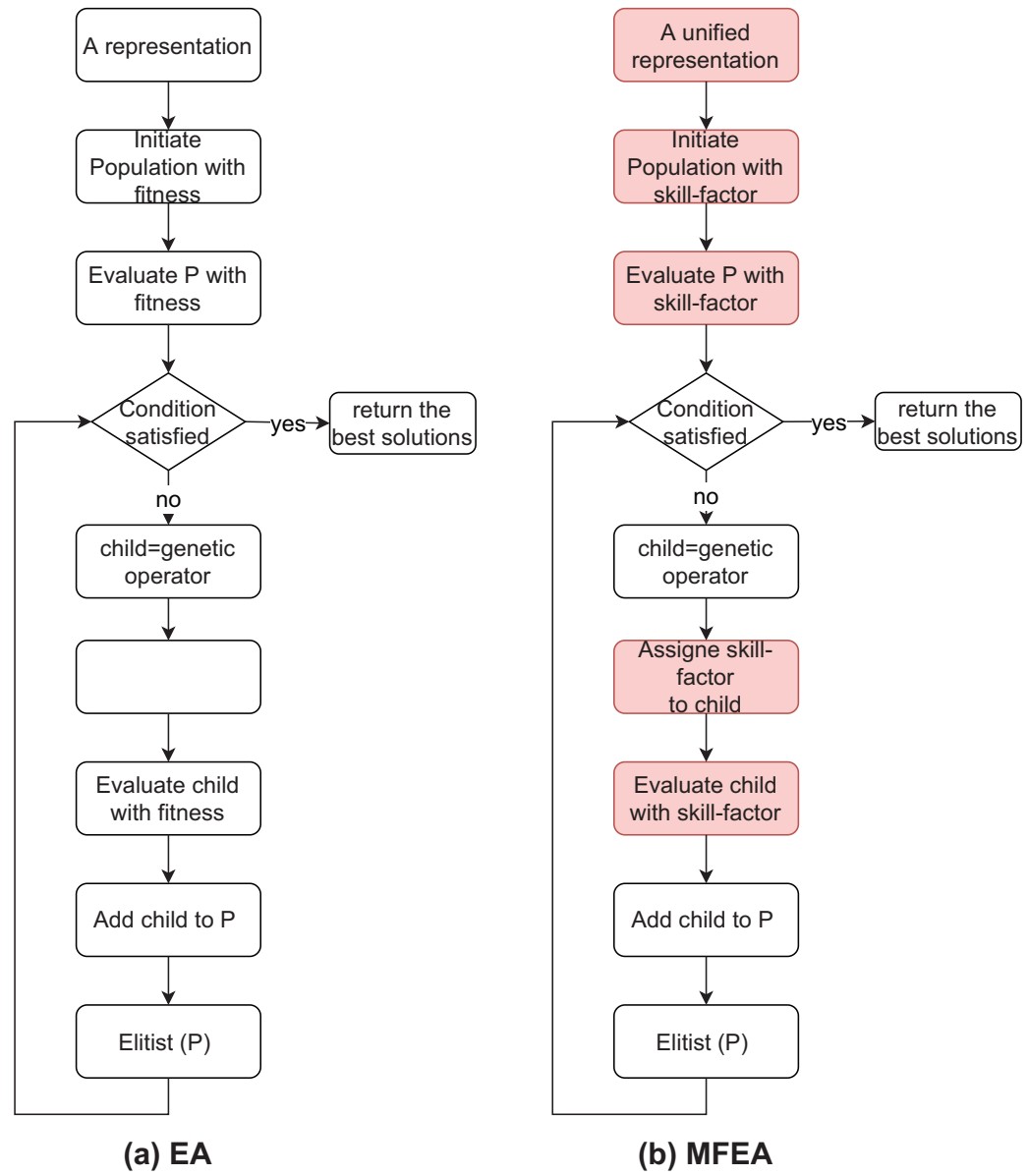

**(a) EA**      **(b) MFEA**

**Figure 1 The similarity and difference between EA and MFEA.**

The MFEA is also unlike multiobjective optimization. In multiobjective optimization, we have one problem with many objective functions. On the other hand, the MFEA solves many tasks at the same time. In addition, multiobjective optimization generally uses a single representative space, while the MFEA unifies multiple representative spaces for many tasks.

Running two algorithms for two tasks independently is not the idea of the MFEA approach. When two algorithms run independently, each task is represented by its own search space. There is no transferrable knowledge between tasks. Otherwise, in the MFEA, two tasks use the unified search space, and transferrable knowledge between tasks is done.

It can increase convergence and improve the quality of solutions for multitasks. *Lian et al. (2019)* then provided a novel theoretical analysis and evidence of the efficiency of the MFEA. This study theoretically explains why some existing the MFEAs perform better than traditional EAs. In addition, the MFEA also can be useful in a system with limited computation.

## THE PROPOSED ALGORITHM

This section introduces the pseudocode of the proposed MFEA+RNVS. The TSPTW task corresponds to a particular task in the MFEA, while another is the TRPTW task. The flow of the proposed algorithm is described in Fig. 2. Our MFEA+RNVS has core components: unified representation, assortative mating (crossover and mutation operators), selective evaluation, scalar-fitness-based selection, RVNS, and Elitism. The detail of the algorithm is shown in Algorithm 2. More specifically, the algorithm includes the following steps. In the first step, a unified search space is created for both two problems. The population with *SP* individuals is then generated in the second step. All solutions for the population must be feasible. After that, the iteration begins until the termination criterion is satisfied. Parents are selected to produce offsprings using crossover or mutation and then assign skill-factors to them. The offsprings are then added to the current population. The individuals of the population are evaluated to update their scalar-fitness and skill-factor. We select the best solutions regarding skill-factor from the current population and convert them from the unified representation to each task's one. It is then fed into the RVNS step to find the best solution for each task. The output of the RVNS is then converted to the unified search space. Finally, it is added to the population. The Elitist strategy keeps the *SP* solutions with the best skill-factors for the next generation.

### Creating unified search space-USS

In the literature, various representations are proposed for two problems. Among these representations, the permutation representation shows efficiency compared to the others. In the permutation, each individual is coded as a set of $n$ vertices $(v_1, v_2, \ldots, v_k, \ldots, v_n)$, where $k$ is the $k-$th index. Figure 3 demonstrates the encoding for two problems.

### Initializing population

Each feasible solution is created from the RVNS to take a role as an individual in the population. Therefore, we have $S_p$ individuals in the initial population for the genetic step.

Algorithm 3 describes the constructive step. The objective function is the sum of all positive differences between the arrival time $(D(v_i))$ and its due time $(l_i)$ on each vertex. Specifically, it is $\min \sum_{i=1}^{n} \max(0, D(v_i) - l_i)$. The algorithm runs until it finds a feasible solution. Restricted Candidate List $(RCL)$ is created by ordering all non-selected vertices based on a greedy function that evaluates the benefit of including them in the solution. One vertex is then chosen from $RCL$ randomly. Since all vertices are visited, we receive a solution. If this solution is a feasible one, it is an initial solution, and this step stops. Conversely, a repair procedure based on the RVNS with many neighborhoods (*Johnson &*

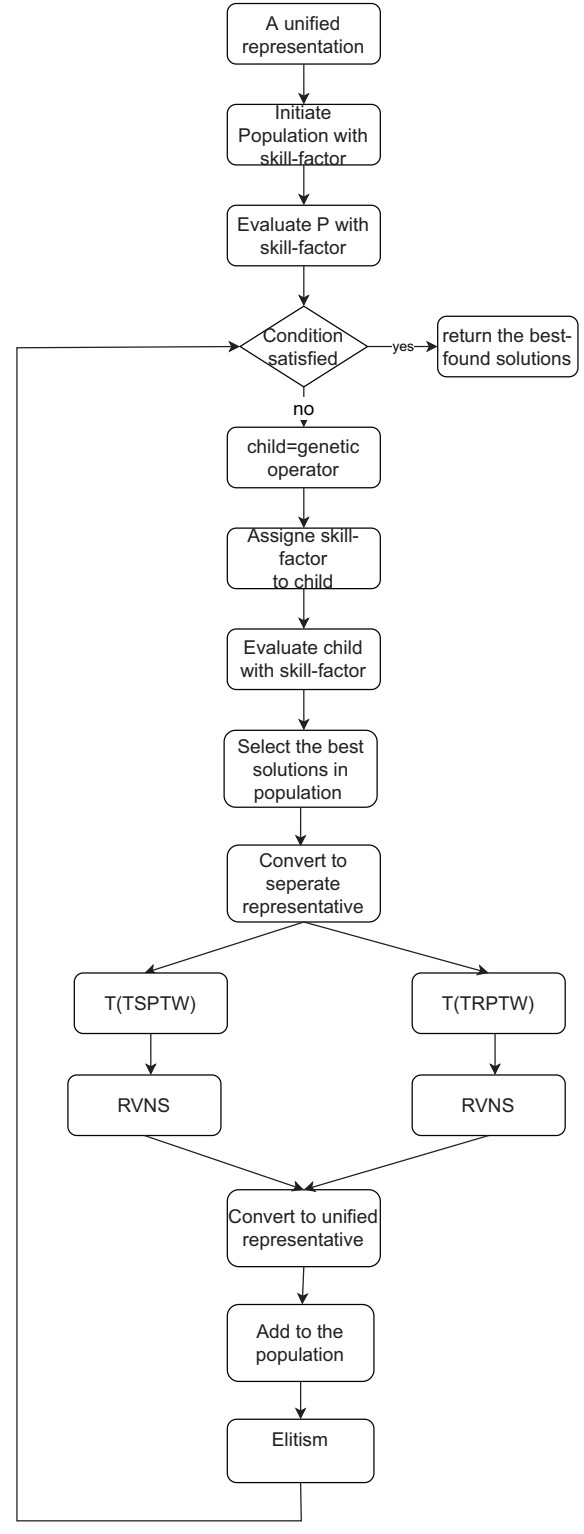

**Figure 2** **The flow of the proposed MFEA+RVNS.**

**Algorithm 2 MFEA+RVNS.**

**Require:** $K_n, C_{ij}, v_1, SP$ are the graph, the cost matrix, the starting vertex, and the size of the population.

**Ensure:** The best solution $T^*_{TSPTW}, T^*_{TRPTW}$.

1: $T^*_{TSPTW}, T^*_{TRPTW} \rightarrow Inf$; {Initiate the best solution for the TSPTW, TRPTW}

2: $P = $ Construction$(v_1, V, k, \alpha, level)$; {Initiate the populatin}

3: **while** (The termination criterion of the MFEA is not satisfied) **do**

4:   {MFEA step (exploration)}

5:   **for** $(j = 1; j \leqslant SP; j++)$**do**

6:     $(P, M) = $ Selection$(P, NG)$; {select parents to mate}

7:     **if** ($M$ and $P$ have the same skill-factor) or (rand(1) $\leqslant rmp$) **then**

8:       **if** ($M$ and $P$ have the same skill-factor) **then**

9:         $C_1, C_2 = $ Crossover$(P, M)$;

10:         $C_1, C_2$'s skill-factors are set to the skill-factors off $P$ or $M$, respectively;

11:       **else if** (rand(1) $\leqslant rmp$) **then**

12:         $C_1, C_2 = $ Crossover$(P, M)$;

13:         $C_1, C_2$'s skill-factors are set to the skill-factors off $P$ or $M$ randomly;

14:       **if** ($C_1$ or $C_2$ is infeasible)**then**

15:         **if** $C_1$ is infeasible **then**

16:           $C_1 = $ Repair$(C_1)$; {convert it to feasible one}

17:         **if** $C_2$ is infeasible **then**

18:           $C_2 = $ Repair$(C_2)$; {convert it to feasible one}

19:   **else**

20:     $C_1 = $ Mutate$(P)$;

21:     $C_2 = $ Mutate$(M)$;

22:     **if** ($C_1$ or $C_2$ isinfeasible) **then**

23:       **if** $C_1$ is infeasible **then**

24:         $C_1 = $ Repair$(C_1)$; {convert it to feasible one}

25:       **if** $C_2$ is infeasible **then**

26:         $C_2 = $ Repair$(C_2)$; {convert it to feasible one}

27:     $C_1$'s, $C_2$'s skill-factor is set to $P, M$, respectively;

28:     $P = P \cup \{C_1, C_2\}$;

29:   Update scalar-fitness and skill-factor for all individuals in $P$;

30:   $LT = $ Select the best individuals from $P$;

31:   {RVNS step (exploitation)}

32:   **for** each $T$ in $LT$ **do**

33:     $(T_{TSPTW}, T_{TRPTW}) = $ Convert $T$ from unified representation to one for each task;

34:     $T'_{TSPTW} = $ RVNS$(T_{TSPTW})$; {local search}

35:     **if** ($T'_{TSPTW} < T^*_{TSPTW}$) **then**

**Algorithm 2 (continued)**

36:      $T'_{TSPTW} \rightarrow T^*_{TSPTW}$;

37:      $T'_{TSPTW} = \text{RVNS}(T_{TSPTW})$; {local search}

38:     **if** $(T'_{TSPTW} < T^*_{TSPTW})$ **then**

39:       $T'_{TSPTW} \rightarrow T^*_{TSPTW}$;

40:      $T' = \text{convert}(T'_{TSPTW}, T'_{TRPTW})$ to unified representation;

41:      $P = P \cup \{T'\}$;

42:    $P = \text{Elitism-Selection}(P)$; {keep the best $SP$ individuals}

43: **return** $T^*_{TSPTW}, T^*_{TRPTW}$;

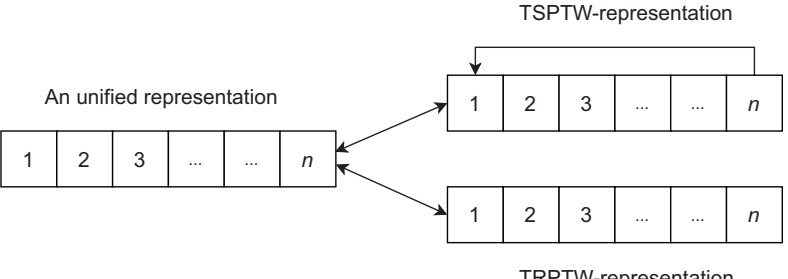

**Figure 3 The interpretation of unified representation for each task.**

---

**Algorithm 3 Construction.**

**Require:** $v_1, V, k, \alpha$, *level* are a starting vertex, the set of vertices in $K_n$, the number of vehicles and the size of *RCL*, the parameter to control the strength of the perturbation procedure, respectively.

**Ensure:** An initial solution $T$.

1: $P = \phi$; {Initially, the population is empty}

2: **while** $(|P| < SP)$ **do**

3:   $T = \{v_1\}$; {$T$ is a tour and it starts at $v_1$}

4:   **while** $|T| < n$ **do**

5:     Create $RCL$ that includes $\alpha$ nearest vertices to $v_e$ in $V$; {$v_e$ is the last vertex in $T$}

6:     Select randomly vertex $v = \{v_i | v_i \in RCL \text{ and } v_i \notin T\}$;

7:     $T \leftarrow T \cup \{v_i\}$; {add the vertex to the tour}

8:   **if** $T$ is infeasible solution **then**

9:     {Convert infeasible solution to feasible one}

10:     $T = \text{Repair}(T, level\_max, N_i(i = 1,\dots,7))$;

11:  $P = P \cup \{T'\}$; {add the tour to the population}

12: return $P$;

---

**Algorithm 4** Repair (*T level_max,$N_i(I = 1,…,7)$*).

**Require:** *T, level_max,$N_i(i = 1,…,7)$* are the infeasible solution, the parameter to control the strength of the perturbation procedure, and the number of neighbourhood respectively.

**Ensure:** An feasible solution *T*.

  1: *level* = 1;
  2: **while** ((*T* is infeasible solution) and (*level* ≤ *level_max*)) **do**
  3:   $T' = \text{Perturbation}(T, level)$;
  4:   **for** $i : 1 \rightarrow 6$ **do**
  5:     $T'' \leftarrow \arg\min N_i(T')$; {local search}
  6:     **if** $(L(T'' < L(T'))$ **then**
  7:       $T' \leftarrow T''$
  8:       $i \leftarrow 1$
  9:     **else**
10:       $i + +$
11:   **if** $L(T') < L(T)$ **then**
12:     $T \leftarrow T'$;
13:   **if** $L(T') == L(T)$ **then**
14:     $level \leftarrow 1$;
15:   **else**
16:     $level + +$;
17: **return** *T*;

---

*McGeoch, 2003*) is invoked, and the procedure iterates until a feasible solution is reached. The solution is shaken to escape from the current local optimal solution. The RVNS is then applied to create the new solution. If it is better than the best-found solution, it is set to the new current solution. The *level* is increased by one if the current solution is not improved, or reset to 1, otherwise. The repair procedure is described in Algorithm 4.

In this article, some neighborhoods widely applied in the literature (*Johnson & McGeoch, 2003*). We describe more details about seven neighborhoods as follows:

- **move** moves a vertex forward one position in *T*.
- **shift** relocates a vertex to another position in *T*.
- **swap-adjacent** attempts to swap each pair of adjacent vertices in the tour.
- **exchange** tries to swap the positions of each pair of vertices in the tour.
- **2-opt** removes each pair of edges from the tour and reconnects them.
- **Or-opt**: Three adjacent vertices are reallocated to another position of the tour.

---

**Algorithm 5** Selection operator (P, NG).

**Require:** *P, NG* are the population and the size of group, respectively.

**Ensure:** Parents $C_1, C_2$.

    1: Select randomly the NG individuals in the *P*;

    2: Sort them in terms of their *R* values;

    3: $C_1, C_2$ = Select two individuals with the best *R* values;

    4: return $C_1, C_2$;

---

## Evaluating for individuals

The scalar-fitness function demonstrates the way of evaluating individuals. Scalar-fitness then are calculated for each individual. The larger and larger the scalar-fitness value is, the better and better the individual is.

## Selection operator

In the original tournament (*Ban & Pham, 2022*; *Talbi, 2009*), the fitness is the only criterion in choosing parents. This article adapts this selection for the MFEA algorithm balancing scalar-fitness and population diversity. It means the selection mechanism simultaneously promoting both diversity and scalar-fitness. For each solution, we count its scalar-fitness and its diversity in a set of solutions as follows:

$$R(T) = (1 - \alpha) \times (SP - RF(T) + 1) + \alpha \times (SP - RD(T) + 1) \tag{1}$$

where $SP$, $\alpha \in [0, 1]$, $RF(T)$, and $RD(T)$ are the population size, threshold, the rank of $T$ in the *P* based on the scalar-fitness, and the rank of $T$ in the *P* based on its diversity, respectively.

$$\bar{d}(T) = \frac{\sum_{i=1}^{n} d(T, T_i)}{n} \tag{2}$$

The metric distance between two solutions is the minimum number of transformations from one to another. We define the distance $d(T, T_i)$ to be $n$ (the number of vertices) minus the number of vertices with the same position on $T$ and $T_i$. Similarly, $\overline{d(T)}$ is the average distance of $T$ in the population. The larger $\bar{d}(T)$ is, the higher its rank is. The larger $R$ is, the better solution $T$ is.

The selection operator selects individual parents based on their $R$ values to mate. We choose the tournament selection operator (*Talbi, 2009*) because of its efficiency. A group of *NG* individuals is selected randomly from the population. Then, two individuals with the best $R$ values in the group are chosen to become parents. The selection pressure can be increased by extending the size of the group. On average, the selected individuals from a larger group have higher $R$ values than those of a small size. The detail in this step is described in Algorithm 5.

## Crossover operator

The crossover is implemented with the predefined probability (*rmp*) or if the parents have the same skill-factor. When parents have the same skill-factor, we have inter-crossover. Otherwise, the intra-crossover is applied to parents with different skill-factor. It opens up the chance for knowledge transfer by using crossover-based exchange between tasks. In *Otman & Jaafar (2011)*, the crossovers are divided into three main types. We found no logical investigation showing which operator brings the best performance in the literature. In a preliminary study, we realize that the algorithm's effectiveness relatively depends on selected crossover operators. Since trying all operators leads to computationally expensive efforts, our numerical analysis is conducted on randomly selected operators for each type. The following operators are selected from the study to balance solution quality and running time.

- The first type is related to the position of certain genes in parents (PMX, CX).
- The second selects genes alternately from both parents, without genes' repetition (EXX, EAX).
- The third is an order-based crossover (SC, MC).

Initially, we select a crossover randomly. If any improvement of the best solution is found, the current crossover operator is continued to use. Otherwise, if the improvement of the best solution is not found after the number of generations (*NO*), another crossover operator is replaced randomly. Using multiple crossovers helps the population be more diverse than one crossover. Therefore, these operators prevent the algorithm from premature convergence. If the offsprings are infeasible, the fix procedure is invoked to convert them to feasible ones. The offsprings' skill-factors are randomly set to the one of the father or mother. The detail in this step is described in Algorithm 6.

## Mutation operator

A mutation is used to keep the diversity of the population. Some mutations are used in the proposed algorithm:

- The Inversion Mutation picks two vertices at random and then inverts the list of vertices between them. It preserves most adjacency information and only breaks two links, leading to the disruption of order information.
- The Insertion Mutation removes the vertex from the current index and then inserts it in a random index on the solution. The operator preserves most of the order and the adjacency information.
- Swap Mutation selects two vertices at random and swaps their positions.

It preserves most of the adjacency information, but links broken disrupt order more. We randomly select one of three operators when this mutation is performed. After the mutation operator, two offsprings are created from the parents. If the offsprings are infeasible, the repair procedure converts them to feasible ones. Their skill-factors are set to those of parents, respectively. The detail in the mutation is described in Algorithm 7.

**Algorithm 6** Crossover (P, M).

**Require:** *P, M* are the parent tour, respectively.

**Ensure:** A new child *T*.

  1: *type* = rand (3);

  2: *rnd* = rand (2);

  3: **if** (*type == 1*) **then**

  4:    {the first type crossover is selected}

  5:    **if** (*rnd == 1*) **then**

  6:        *C* = **PMX** (*P, M*); {PMX is chosen}

  7:    **else if** (*rnd==2*) **then**

  8:        *C* = **CX** (*P, M*); {CX is selected}

  9: **else if** (*type==2*) **then**

10:    {the second type is selected}

11:    **if** (*rnd == 1*) **then**

12:        *C* = **EXX** (*P, M*); {EXX is selected}

13:    **else if** (*rnd==2*) **then**

14:        *C* = **EAX** (*P, M*); {EAX is selected}

15: **else if** (*type==3*) **then**

16:    {the type 3 is selected}

17:    **if** (*rnd == 1*) **then**

18:        *C* = **SC** (*P, M*); {SC is selected}

19:    **else if** (*rnd==2*) **then**

20:        *C* = **MC** (*P, M*); {MC is selected}

## RVNS

The combination between the MFEA and the RVNS allows good transferrable knowledge between tasks from the MFEA and the ability to exploit good solution spaces from RVNS. We select some best solutions in the current population to feed into the RVNS. In the RVNS step, we convert a solution from unified representation to separated representation for each task. The RVNS then applies to each task separately. Finally, the output of the RVNS is represented in the unified space. The improved solution will be added to the population.

For this step, we use popular neighborhoods such as move, shift, swap-adjacent, exchange, 2-opt, and or-opt in *Johnson & McGeoch (2003)* and *Reeves (1999)*. In addition, the pseudocode of the RVNS algorithm is given in Algorithm 8.

## Elitism operator

Elitism is a process that ensures the survival of the fittest, so they do not die through the evolutionary processes. Researchers show the number (*Talbi, 2009*) (usually below 15%) of

---

**Algorithm 7** Mutate (*C*).

**Require:** *C* is the child, respectively.

**Ensure:** A new child *C*.

    1: {Choose a mutation operator randomly}

    2: *rnd* = rand (2);

    3: **if** (*rnd == 1*) **then**

    4:     *C* = **Inversion(C)** {Inversion mutation is selected}

    5: **else if** (*rnd==2*) **then**

    6:     *C* = **Insertion(C)** {Inversion mutation is selected}

    7: **else**

    8:     *C* = **Swap(C)** {Swap mutation is selected}

    9: return *C*;

---

**Algorithm 8** RVNS (T).

**Require:** *T* is a tour.

**Ensure:** A new solution *C*.

    1: Initialize the Neighbourhood List *NL*;

    2: **while** $NL \neq 0$ **do**

    3:   Choose a neighbourhood *N* in *NL* at random

    4:     $T' \leftarrow \arg\min N(T)$; {Neighbourhood search}

    5:     **if** $((W(T') < W(T))$ and $(T'$ isfeasible)) **then**

    6:         $T \leftarrow T'$

    7:         Update *NL*;

    8:     **else**

    9:         Remove *N* from the *NL*;

    10:   **if** $((W(T') < W(T^*))$ and $(T'$ isfeasible)) **then**

    11:     $T^* \leftarrow T'$

---

the best solutions that automatically go to the next generation. The proposed algorithm selects *Sp* individuals for the next generation, in which about 15% of them are the best solutions in the previous generation, and the remaining individuals are chosen randomly from *P*.

## The stop condition

After the number of generations (*Ng*), if the best solution has not been improved, then the proposed algorithm stops.

**Table 1 The variable parameters.**

| Parameter | Value range |
|---|---|
| SP | $50 \leq \beta_r \leq 200$, incremented by 50 |
| NG | $5 \leq \alpha \leq 15$, incremented by 5 |
| rmp | $0.5 \leq \beta_\eta \leq 1$, incremented by 0.1 |
| α | $5 \leq \tau_0 \leq 20$, incremented by 5 |
| level | $5 \leq p \leq 15$, incremented by 5 |
| Ng | $50 \leq Ng \leq 150$, incremeted by 50 |

## COMPUTATIONAL EVALUVATIONS

The experiments are conducted on a personal computer equipped with a Xeon E-2234 CPU and 16 GB bytes of RAM. The program was coded in C≠ language. The generation number ($Ng$), population size ($SP$), group size ($NG$) in the selection, and crossover rates ($rmp$) influence the algorithm's results. Many efforts in the literature studied the algorithm sensitivity to parameter changes. We found that no work shows which values are the best for all cases. However, the following suggestions help us in choosing parameter values:

- A large generation number does not improve performance. Besides, it consumes much time to run. A small value makes the algorithm fail to reach the best solution (*Angelova & Pencheva, 2011*).
- A higher crossover value obtains new individuals more quickly while a low crossover rate may cause stagnation (*Angelova & Pencheva, 2011*).
- A large population size can increase the population diversity. However, it can be unhelpful in the algorithm and increase the running time of it (*Chen et al., 2012*).
- Increased selection pressure can be provided by simply increasing the group size. When the selection pressure is too low, the convergence rate is slower, while if it is too high, the chance of the algorithm prematurely converges (*Lavinas et al., 2019*).
- The α and level values help to create the diversity of the initial population. A larger value leads to the same as the random method, while a small value decreases the diversity.

Based on the suggestions, we determine a suitable range for each parameter in Table 1. In the next step, we choose the best value from the range as follows: finding the best configuration by conducting all instances would have been too expensive in computation, and we test numerical analysis on some instances. The configuration selected in many combinations is tested, and the one that has obtained the best solution is chosen. In Table 1, we determine a range for each parameter that generates different combinations, and we run the proposed algorithm on some selected instances of the combinations. We find the following settings so that our algorithm obtains the best solutions: $SP = 100, NG = 5, rmp = 0.7, \alpha = 10, level = 5$, and $Ng = 100$. This parameter setting has thus been used in the following experiments.

We found no algorithm based on the literature's MFEA framework for the TRPTW and TSPTW. Therefore, the proposed algorithm's results directly compare with the known best solutions of the TSPTW and TRPTW on the same benchmark. Moreover, to compare with the previous MFEA framework (*Osaba et al., 2020*; *Yuan et al., 2016*), our MEFA+RVNS is tested on the benchmark for the TSP and TRP. They are specific variants of TSPTW and TRPTW without time window constraints. Therefore, the instances are used in the article as follows:

- *Dumas, Desrosiers & Gélinas (1995)* propose the first set citebib09 and contains 135 instances grouped in 27 test cases. Each group has five Euclidean instances, coordinates between 0 and 50, with the same number of customers and the same maximum range of time windows. For example, the instances n20w60.001, n20w60.002, n20w60.003, n20w60.004, and n20w60.005 have 20 vertices and the time window for each vertex is uniformly random, between 0 and 60.
- *Gendreau et al. (1998)* propose the second set of instances citebib12 and contains 140 instances grouped in 28 test cases.
- *Ohlmann & Thomas (2007)* propose the third set of instances citebib30 and contains 25 instances grouped in five test cases.
- The fourth sets in the majority are the instances proposed by *Dumas, Desrosiers & Gélinas (1995)* with wider time windows.
- The TSPLIB (http://comopt.ifi.uni-heidelberg.de/software/TSPLIB95/) includes fourteen instances from 50 to 100 instances.

The efficiency of the metaheuristic algorithm can be evaluated by comparing the best solution found by the proposed algorithm (notation: *Best.Sol*) to (1) the optimal solution (notation: *OPT*); and (2) the known best solution (notation: *KBS*) of the previous metaheuristics (note that: In the TSPTW, *KBS* is the optimal solution) as follows:

$$gap[\%] = \frac{Best.Sol - KBS(OPT)}{KBS(OPT)} \times 100\% \qquad (3)$$

The smaller and smaller the value of *gap* is, the better and better our solution is. All instances and found solutions are available in the link https://sites.google.com/soict.hust.edu.vn/mfea-tsptw-trptw/home.

In Tables, *OPT*, *Aver.Sol* and *Best.Sol* are the optimal, average, and best solution after ten runs, respectively. Let *Time* be the running time such that the proposed algorithm reaches the best solution. Note that: *Yuan et al. (2016)* supported the source code of their algorithm in *Yuan et al. (2016)* while the dMFEA-II (dMFEA-II is the MFEA with dynamic *rmp* value) (*Osaba et al., 2020*) was implemented again by us. Tables 2 and 3 evaluate the efficiency of the proposed selection in MFEA+RVNS. Tables 4–8 compare the proposed MFEA+RVNS with the known best or optimal solutions for the TSPTW and TRPTW instances (*Abeledo et al., 2013*; *Ban & Nghia, 2017*; *Ban, 2021*; *Dumas, Desrosiers & Gélinas, 1995*; *Heilporna, Cordeaua & Laporte, 2010*; *Silva & Urrutia, 2010*; *Ohlmann & Thomas, 2007*; *Salvesbergh, 1985*; *Reeves, 1999*). In the Tables, the *KBS*, *OPT*, *Aver.Sol*, and

**Table 2 Comparison the best-found values between MFEA-NR and MFEA for TSPTW and TRPTW instances proposed by _Dumas, Desrosiers & Gélinas (1995)_, and _Silva & Urrutia (2010)_.**

| Instances | MFEA-NR | | MFEA+RNVS | | diff[%] | |
|---|---|---|---|---|---|---|
| | TSPTW | TRPTW | TSPTW | TRPTW | TSPTW | TRPTW |
| n20w20.002 | 286 | 2,560 | 286 | 2,560 | 0.00 | 0.00 |
| n20w40.002 | 333 | 2,679 | 333 | 2,679 | 0.00 | 0.00 |
| n20w60.002 | 244 | 2,176 | 244 | 2,176 | 0.00 | 0.00 |
| n20w80.003 | 338 | 2,669 | 338 | 2,669 | 0.00 | 0.00 |
| n20w100.002 | 222 | 2,082 | 222 | 2,082 | 0.00 | 0.00 |
| n40w40.002 | 483 | 7,202 | 461 | 7,104 | −4.55 | −1.36 |
| n40w60.002 | 487 | 7,303 | 470 | 7,247 | −3.49 | −0.77 |
| n40w80.002 | 468 | 7,209 | 431 | 7,123 | −7.91 | −1.19 |
| n40w100.002 | 378 | 6,789 | 364 | 6,693 | −3.70 | −1.41 |
| n60w20.002 | 626 | 14,003 | 605 | 13,996 | −3.35 | −0.05 |
| n60w120.002 | 472 | 12,622 | 427 | 12,525 | −9.53 | −0.77 |
| n60w140.002 | 475 | 11,914 | 464 | 11,810 | −2.32 | −0.87 |
| n60w160.002 | 443 | 12,920 | 423 | 12,719 | −4.51 | −1.56 |
| n80w120.002 | 587 | 18,449 | 577 | 18,383 | −1.70 | −0.36 |
| n80w140.002 | 495 | 18,243 | 472 | 18,208 | −4.65 | −0.19 |
| n80w160.002 | 588 | 17,334 | 553 | 17,200 | −5.95 | −0.77 |
| aver | | | | | −3.23 | −0.58 |

_Best.Sol_ columns are the best known, optimal, average, and best solution, respectively, while the _gap_ column presents the difference between the best solution and the optimal one. Table 9 shows the average values of Tables 4–7 comparing the MFEA+RVNS, OA (_Osaba et al., 2020_), and YA (_Yuan et al., 2016_). In the TSP, the optimal solutions of the TSPLIB-instances are obtained by running the Concord tool (https://www.math.uwaterloo.ca/tsp/concorde.html). The optimal or best solutions in the TRP are obtained from _Abeledo et al. (2013)_.

## Evaluating the efficiency of selection

In this experiment, a new selection operator for the MFEA+RVNS algorithm effectively balances knowledge transfer and diversity. Due to being too expensive in computation, we choose some instances to evaluate the efficiency of this operator. In Table 2, the column MFEA-NR results from the MFEA+RVNS with the selection-based scalar-fitness only, while column MFEA+RVNS is the results of the MFEA+RVNS with both scalar-fitness and diversity. The _diff_ [%] column is the difference between the MFEA+RVNS and MFEA-NR in percentage.

In Table 2, the MFEA+RVNS outperforms the MFEA-NR in all cases. The selection operation that considers both scalar-fitness and diversity to pick parents is more effective than the one with scalar-fitness only. The fitness-based criterion promotes "survival of the fittest", which is good for transferring elite genes between tasks, and good individuals are

**Table 3  Comparison the best-found values between MFEA-NLS and MFEA for TSPTW and TRPTW instances proposed by** *Dumas, Desrosiers & Gélinas (1995)*, **and** *Silva & Urrutia (2010)*.

| Instances | MFEA-NLS | | MFEA+RNVS | | diff[%] | |
|---|---|---|---|---|---|---|
| | TSPTW | TRPTW | TSPTW | TRPTW | TSPTW | TRPTW |
| n20w20.002 | 286 | 2,560 | 286 | 2,560 | 0.00 | 0.00 |
| n20w40.002 | 333 | 2,679 | 333 | 2,679 | 0.00 | 0.00 |
| n20w60.002 | 244 | 2,176 | 244 | 2,176 | 0.00 | 0.00 |
| n20w80.003 | 338 | 2,669 | 338 | 2,669 | 0.00 | 0.00 |
| n20w100.002 | 222 | 2,082 | 222 | 2,082 | 0.00 | 0.00 |
| n40w40.002 | 522 | 7,530 | 461 | 7,104 | −11.69 | −5.66 |
| n40w60.002 | 503 | 7,517 | 470 | 7,247 | −6.56 | −3.59 |
| n40w80.002 | 475 | 7,763 | 431 | 7,123 | −9.26 | −8.24 |
| n40w100.002 | 400 | 7,502 | 364 | 6,693 | −9.00 | −10.78 |
| n60w20.002 | 626 | 14,097 | 605 | 13,996 | −3.35 | −0.72 |
| n60w120.002 | 480 | 13,680 | 427 | 12,525 | −11.04 | −8.44 |
| n60w140.002 | 502 | 12,951 | 464 | 11,810 | −7.57 | −8.81 |
| n60w160.002 | 461 | 13,953 | 423 | 12,719 | −8.24 | −8.84 |
| n80w120.002 | 620 | 19,860 | 577 | 18,383 | −6.94 | −7.44 |
| n80w140.002 | 520 | 19,742 | 472 | 18,208 | −9.23 | −7.77 |
| n80w160.002 | 614 | 19,516 | 553 | 17,200 | −9.93 | −11.87 |
| aver | | | | | −5.80 | −5.14 |

**Table 4  Comparison between our results with the best-found values for TSPTW and TRPTW instances proposed by** *Dumas, Desrosiers & Gélinas (1995)*, **and** *Silva & Urrutia (2010)*.

| Instances | TSPTW | TRPTW | MFEA+RNVS | | | | | | | |
|---|---|---|---|---|---|---|---|---|---|---|
| | | | TSPTW | | | | TRPTW | | | |
| | OPT | KBS | Best.Sol | Aver.Sol | gap | Time | Best.Sol | Aver.Sol | gap | Time |
| n20w20.001 | 378 | 2,528 | 378 | 378 | 0.0 | 3 | 2,528 | 2,528 | 0.0 | 3 |
| n20w20.002 | 286 | 2,560 | 286 | 286 | 0.0 | 2 | 2,560 | 2,560 | 0.0 | 2 |
| n20w20.003 | 394 | 2,671 | 394 | 394 | 0.0 | 2 | 2,671 | 2,671 | 0.0 | 2 |
| n20w20.004 | 396 | 2,975 | 396 | 396 | 0.0 | 6 | 2,975 | 2,975 | 0.0 | 6 |
| n20w40.001 | 254 | 2,270 | 254 | 254 | 0.0 | 2 | 2,270 | 2,270 | 0.0 | 2 |
| n20w40.002 | 333 | 2,679 | 333 | 333 | 0.0 | 5 | 2,679 | 2,679 | 0.0 | 5 |
| n20w40.003 | 317 | 2,774 | 317 | 317 | 0.0 | 3 | 2,774 | 2,774 | 0.0 | 2 |
| n20w40.004 | 388 | 2,568 | 388 | 388 | 0.0 | 2 | 2,568 | 2,568 | 0.0 | 3 |
| n20w60.001 | 335 | 2,421 | 335 | 335 | 0.0 | 3 | 2,421 | 2,421 | 0.0 | 2 |
| n20w60.002 | 244 | 2,176 | 244 | 244 | 0.0 | 2 | 2,176 | 2,176 | 0.0 | 2 |
| n20w60.003 | 352 | 2,694 | 352 | 352 | 0.0 | 2 | 2,694 | 2,694 | 0.0 | 2 |
| n20w60.004 | 280 | 2,020 | 280 | 280 | 0.0 | 2 | 2,020 | 2,020 | 0.0 | 2 |
| n20w80.001 | 329 | 2,990 | 329 | 329 | 0.0 | 2 | 2,990 | 2,990 | 0.0 | 3 |
| n20w80.002 | 338 | 2,669 | 340 | 340 | 0.6 | 1 | 2,669 | 2,669 | 0.0 | 2 |

| Instances | TSPTW | TRPTW | MFEA+RNVS | | | | | | | | |
|---|---|---|---|---|---|---|---|---|---|---|---|
| | | | TSPTW | | | | TRPTW | | | |
| | OPT | KBS | Best.Sol | Aver.Sol | gap | Time | Best.Sol | Aver.Sol | gap | Time |
| n20w80.003 | 320 | 2,643 | 320 | 320 | 0.0 | 2 | 2,643 | 2,643 | 0.0 | 2 |
| n20w80.004 | 304 | 2,627 | 306 | 306 | 0.7 | 3 | 2,552 | 2,552 | −2.9 | 2 |
| n20w100.001 | 237 | 2,294 | 237 | 237 | 0.0 | 3 | 2,269 | 2,269 | −1.1 | 3 |
| n20w100.002 | 222 | 2,082 | 222 | 222 | 0.0 | 2 | 2,082 | 2,082 | 0.0 | 2 |
| n20w100.003 | 310 | 2,416 | 310 | 310 | 0.0 | 2 | 2,416 | 2,416 | 0.0 | 3 |
| n20w100.004 | 349 | 2,914 | 349 | 349 | 0.0 | 1 | 2,862 | 2,862 | −1.8 | 2 |
| n40w20.001 | 500 | 7,875 | 500 | 500 | 0.0 | 9 | 7,875 | 7,875 | 0.0 | 8 |
| n40w20.002 | 552 | 7,527 | 552 | 552 | 0.0 | 7 | 7,527 | 7,527 | 0.0 | 8 |
| n40w20.003 | 478 | 7,535 | 478 | 478 | 0.0 | 8 | 7,535 | 7,535 | 0.0 | 9 |
| n40w20.004 | 404 | 7,031 | 404 | 404 | 0.0 | 8 | 7,031 | 7,031 | 0.0 | 9 |
| n40w40.001 | 465 | 7,663 | 465 | 465 | 0.0 | 7 | 7,663 | 7,663 | 0.0 | 9 |
| n40w40.002 | 461 | 7,104 | 461 | 461 | 0.0 | 8 | 7,104 | 7,104 | 0.0 | 8 |
| n40w40.003 | 474 | 7,483 | 474 | 474 | 0.0 | 8 | 7,483 | 7,483 | 0.0 | 8 |
| n40w40.004 | 452 | 6,917 | 452 | 452 | 0.0 | 8 | 6,917 | 6,917 | 0.0 | 9 |
| n40w60.001 | 494 | 7,066 | 494 | 494 | 0.0 | 9 | 7,066 | 7,066 | 0.0 | 7 |
| n40w60.002 | 470 | 7,247 | 470 | 470 | 0.0 | 8 | 7,247 | 7,247 | 0.0 | 8 |
| n40w60.003 | 408 | 6,758 | 410 | 410 | 0.0 | 9 | 6,758 | 6,758 | 0.0 | 8 |
| n40w60.004 | 382 | 5,548 | 382 | 382 | 0.0 | 9 | 5,548 | 5,548 | 0.0 | 9 |
| n40w80.001 | 395 | 8,229 | 395 | 395 | 0.0 | 8 | 8,152 | 8,152 | 0.0 | 9 |
| n40w80.002 | 431 | 7,176 | 431 | 431 | 0.0 | 8 | 7,123 | 7,123 | 0.0 | 9 |
| n40w80.003 | 412 | 7,075 | 418 | 418 | 0.0 | 8 | 7,075 | 7,075 | 0.0 | 9 |
| n40w80.004 | 417 | 7,166 | 417 | 417 | 0.6 | 9 | 7,166 | 7,166 | 0.0 | 10 |
| n40w100.001 | 429 | 6,858 | 432 | 432 | 0.0 | 8 | 68,00 | 6,800 | 0.0 | 9 |
| n40w100.002 | 358 | 6,778 | 364 | 364 | 0.7 | 11 | 6,693 | 6,693 | −2.9 | 10 |
| n40w100.003 | 364 | 6,178 | 364 | 364 | 0.0 | 9 | 6,926 | 6,926 | −1.1 | 11 |
| n40w100.004 | 357 | 7,019 | 361 | 361 | 0.0 | 9 | 7,019 | 7,019 | 0.0 | 8 |
| n60w20.002 | 605 | 13,996 | 605 | 605 | 0.0 | 18 | 13,996 | 13,996 | 0.0 | 19 |
| n60w20.003 | 533 | 13,782 | 533 | 533 | 0.0 | 17 | 12,965 | 12,965 | −1.8 | 18 |
| n60w20.004 | 616 | 12,965 | 616 | 616 | 0.0 | 17 | 15,102 | 15,102 | 0.0 | 18 |
| n60w40.003 | 603 | 15,034 | 612 | 612 | 0.0 | 19 | 15,034 | 15,034 | 0.0 | 19 |

kept in each task, implying an accumulation of good genes. However, population diversity is important since it becomes a bottleneck. Our selection ensures that selection recognizes both diversity contribution and fitness in choosing the best individuals for reproduction.

## Evaluating the balance between exploration and exploitation

Generally speaking, algorithms get stuck into local optimum because there is a lack of balance between exploration and exploitation. Exploration helps the search to explore extension spaces on a global scale, while exploitation helps the search to focus on local

**Table 5 Comparison between our results with the best-found values for TSPTW and TRPTW instances proposed by *Dumas, Desrosiers & Gélinas (1995)*, and *Silva & Urrutia (2010)*.**

| Instances | TSPTW | TRPTW | MFEA+RNVS | | | | | | | | |
|---|---|---|---|---|---|---|---|---|---|---|
| | | | TSPTW | | | | TRPTW | | | |
| | OPT/KBS | KBS | Best.Sol | Aver.Sol | Gap | Time | Best.Sol | Aver.Sol | Gap | Time |
| n60w160.004 | 401 | 11,645 | 401 | 401 | 0.0 | 19 | 11,778 | 11,778 | 1.1 | 19 |
| n60w180.002 | 399 | 12,015 | 399 | 399 | 0.0 | 17 | 12,224 | 12,224 | 1.7 | 21 |
| n60w180.003 | 445 | 12,214 | 445 | 445 | 0.0 | 18 | 12,679 | 12,679 | 3.8 | 21 |
| n60w180.004 | 456 | 11,101 | 456 | 456 | 0.0 | 19 | 11,245 | 11,245 | 1.3 | 18 |
| n60w200.002 | 414 | 11,748 | 414 | 414 | 0.0 | 20 | 11,866 | 11,866 | 1.0 | 19 |
| n60w200.003 | 455 | 10,697 | 460 | 460 | 1.1 | 19 | 10,697 | 10,697 | 0.0 | 18 |
| n60w200.004 | 431 | 11,441 | 431 | 431 | 0.0 | 16 | 11,740 | 11,740 | 2.6 | 17 |
| n80w120.002 | 577 | 18,181 | 577 | 577 | 0.0 | 17 | 18,383 | 18,383 | 1.1 | 21 |
| n80w120.003 | 540 | 17,878 | 548 | 548 | 1.5 | 19 | 17,937 | 17,937 | 0.3 | 21 |
| n80w120.004 | 501 | 17,318 | 501 | 501 | 0.0 | 18 | 17,578 | 17,578 | 1.5 | 18 |
| n80w140.002 | 472 | 17,815 | 472 | 472 | 0.0 | 19 | 18,208 | 18,208 | 2.2 | 21 |
| n80w140.003 | 580 | 17,315 | 580 | 580 | 0.0 | 20 | 17,358 | 17,358 | 0.2 | 21 |
| n80w140.004 | 424 | 18,936 | 424 | 424 | 0.0 | 19 | 19,374 | 19,374 | 2.3 | 18 |
| n80w160.002 | 553 | 17,091 | 553 | 553 | 0.0 | 27 | 17,200 | 17,200 | 0.6 | 18 |
| n80w160.003 | 521 | 16,606 | 521 | 521 | 0.0 | 21 | 16,521 | 16,521 | −0.5 | 20 |
| n80w160.004 | 509 | 17,804 | 509 | 509 | 0.0 | 20 | 17,927 | 17,927 | 0.7 | 18 |
| n80w180.002 | 479 | 17,339 | 479 | 479 | 0.0 | 21 | 17,904 | 17,904 | 3.3 | 19 |
| n80w180.003 | 530 | 17,271 | 530 | 530 | 0.0 | 20 | 17,160 | 17,160 | −0.6 | 20 |
| n80w180.004 | 479 | 16,729 | 479 | 479 | 0.0 | 22 | 16,849 | 16,849 | 0.7 | 21 |
| n100w120.002 | 540 | 29,882 | 556 | 556 | 3.0 | 38 | 29,818 | 29,818 | 0.0 | 45 |
| n100w120.003 | 617 | 25,275 | 646 | 646 | 4.7 | 37 | 24,473 | 24,473 | 0.0 | 42 |
| n100w120.004 | 663 | 30,102 | 663 | 663 | 0.0 | 39 | 31,554 | 31,554 | 0.0 | 41 |
| n100w140.002 | 622 | 30,192 | 632 | 632 | 1.6 | 38 | 30,087 | 30,087 | 0.0 | 45 |
| n100w140.003 | 481 | 28309 | 481 | 481 | 0.0 | 39 | 28,791 | 28,791 | 0.0 | 47 |
| n100w140.004 | 533 | 27,448 | 533 | 533 | 0.0 | 40 | 27,990 | 27,990 | 0.0 | 45 |
| n150w120.003 | 747 | 42,340 | 769 | 769 | 2.9 | 75 | 42,339 | 42,339 | 0.0 | 72 |
| n150w140.001 | 762 | 42,405 | 785 | 785 | 3.0 | 70 | 42,388 | 42,388 | −0.1 | 74 |
| n150w160.001 | 706 | 45,366 | 732 | 732 | 3.6 | 72 | 45,160 | 45,160 | −0.4 | 78 |
| n150w160.002 | 711 | 44,123 | 735 | 735 | 3.3 | 74 | 44,123 | 44,123 | 0.0 | 76 |
| n200w200.001 | 9,424 | 1,094,630 | 9,424 | 9,424 | 0.0 | 101 | 1,093,537 | 1,093,537 | −0.1 | 89 |
| n200w200.002 | 9,838 | 1,099,839 | 9,885 | 9,885 | 0.5 | 110 | 1,099,839 | 1,099,839 | 0.0 | 86 |
| n200w200.003 | 9,043 | 1,067,171 | 9,135 | 9,135 | 1.0 | 99 | 1,067,161 | 1,067,161 | 0.0 | 93 |
| n200w300.001 | 7,656 | 1,052,884 | 7,791 | 7,791 | 1.7 | 100 | 1,047,893 | 1,047,893 | −0.5 | 106 |
| n200w300.002 | 7,578 | 1,047,893 | 7,721 | 7,721 | 1.8 | 105 | 1,047,893 | 1,047,893 | 0.0 | 110 |
| n200w300.003 | 8,600 | 1,069,169 | 8,739 | 8,739 | 1.6 | 120 | 1,069,169 | 1,069,169 | 0.0 | 93 |
| n200w300.004 | 8,268 | 1,090,972 | 8,415 | 8,415 | 1.7 | 112 | 1,090,972 | 1,090,972 | 0.0 | 96 |
| n200w300.005 | 8,030 | 1,022,000 | 8,190 | 8,190 | 1.9 | 114 | 1,016,765 | 1,016,765 | −5.1 | 98 |
| n200w400.001 | 7,420 | 1,064,456 | 7,661 | 7,661 | 3.2 | 109 | 1,064,456 | 1,064,456 | 0.0 | 100 |
| aver | | | | | 0.49 | 26.24 | | | −0.11 | 25.6 |

**Table 6 Comparison between our results with the best-found values for TSPTW and TRPTW instances proposed by *Gendreau et al. (1998)*, and *Ohlmann & Thomas (2007)*.**

| Instances | TSPTW | TRPTW | MFEA+RNVS | | | | | | | |
|---|---|---|---|---|---|---|---|---|---|---|
| | | | TSPTW | | | | TRPTW | | | |
| | KBS | KBS | Best.Sol | Aver.Sol | Gap | Time | Best.Sol | Aver.Sol | Gap | Time |
| n20w120.002 | 218 | 2,193 | 218 | 218 | 0.0 | 2 | 2,193 | 2,193 | 0.0 | 3 |
| n20w120.003 | 303 | 2,337 | 303 | 303 | 0.0 | 4 | 2,337 | 2,337 | 0.0 | 2 |
| n20w120.004 | 300 | 2,686 | 300 | 300 | 0.0 | 2 | 2,686 | 2,686 | 0.0 | 2 |
| n20w140.002 | 272 | 2,330 | 272 | 272 | 0.0 | 2 | 2,330 | 2,330 | 0.0 | 3 |
| n20w140.003 | 236 | 2,194 | 236 | 236 | 0.0 | 2 | 2,196 | 2,196 | 0.1 | 2 |
| n20w140.004 | 255 | 2,279 | 264 | 264 | 3.5 | 4 | 2,278 | 2,278 | 0.0 | 5 |
| n20w160.002 | 201 | 1,830 | 201 | 201 | 0.0 | 2 | 1,830 | 1,830 | 0.0 | 2 |
| n20w160.003 | 201 | 2,286 | 201 | 201 | 0.0 | 3 | 2,286 | 2,286 | 0.0 | 2 |
| n20w160.004 | 203 | 1,616 | 203 | 203 | 0.0 | 2 | 1,616 | 1,616 | 0.0 | 2 |
| n20w180.002 | 265 | 2,315 | 265 | 265 | 0.0 | 4 | 2,315 | 2,315 | 0.0 | 2 |
| n20w180.003 | 271 | 2,414 | 271 | 271 | 0.0 | 2 | 2,414 | 2,414 | 0.0 | 2 |
| n20w180.004 | 201 | 2,624 | 201 | 201 | 0.0 | 3 | 1,924 | 1,924 | −26.7 | 2 |
| n20w200.002 | 203 | 1,799 | 203 | 203 | 0.0 | 2 | 1,799 | 1,799 | 0.0 | 2 |
| n20w200.003 | 249 | 2,144 | 260 | 260 | 4.4 | 2 | 2,089 | 2,089 | −2.6 | 1 |
| n20w200.004 | 293 | 2,624 | 293 | 293 | 0.0 | 1 | 2,613 | 2,613 | −0.4 | 2 |
| n40w120.002 | 445 | 6,265 | 446 | 446 | 0.2 | 3 | 6,265 | 6,265 | 0.0 | 8 |
| n40w120.003 | 357 | 6,411 | 360 | 360 | 0.8 | 2 | 6,411 | 6,411 | 0.0 | 7 |
| n40w120.004 | 303 | 5,855 | 303 | 303 | 0.0 | 3 | 5,855 | 5,855 | 0.0 | 6 |
| n40w140.002 | 383 | 5,746 | 383 | 383 | 0.0 | 2 | 5,746 | 5,746 | 0.0 | 8 |
| n40w140.003 | 398 | 6,572 | 398 | 398 | 0.0 | 3 | 6,572 | 6,572 | 0.0 | 7 |
| n40w140.004 | 342 | 5,719 | 350 | 350 | 2.3 | 8 | 5,680 | 5,680 | −0.7 | 8 |
| n40w160.002 | 337 | 6,368 | 338 | 338 | 0.3 | 9 | 6,351 | 6,351 | −0.3 | 8 |
| n40w160.003 | 346 | 5,850 | 346 | 346 | 0.0 | 9 | 5,850 | 5,850 | 0.0 | 9 |
| n40w160.004 | 288 | 4,468 | 289 | 289 | 0.3 | 8 | 4,440 | 4,440 | −0.6 | 9 |
| n40w180.002 | 347 | 6,104 | 349 | 349 | 0.6 | 8 | 6,104 | 6,104 | 0.0 | 9 |
| n40w180.003 | 279 | 6,040 | 282 | 282 | 1.1 | 7 | 6,031 | 6,031 | −0.1 | 8 |
| n40w180.004 | 354 | 6,103 | 361 | 361 | 2.0 | 8 | 6,283 | 6,283 | 2.9 | 8 |
| n40w200.002 | 303 | 6,674 | 303 | 303 | 0.0 | 8 | 5,830 | 5,830 | −12.6 | 9 |
| n40w200.003 | 339 | 5,542 | 343 | 343 | 1.2 | 7 | 5,230 | 5,230 | −5.6 | 8 |
| n40w200.004 | 301 | 6,103 | 301 | 301 | 0.0 | 9 | 5,977 | 5,977 | −2.1 | 10 |
| n60w120.002 | 427 | 12,517 | 427 | 427 | 0.0 | 19 | 12,525 | 12,525 | 0.1 | 19 |
| n60w120.003 | 407 | 11,690 | 419 | 419 | 2.9 | 19 | 11,680 | 11,680 | −0.1 | 19 |
| n60w120.004 | 490 | 11,132 | 492 | 492 | 0.4 | 13 | 11,135 | 11,135 | 0.0 | 19 |
| n60w140.002 | 462 | 11,782 | 464 | 464 | 0.4 | 18 | 11,810 | 11,810 | 0.2 | 19 |
| n60w140.003 | 427 | 13,128 | 448 | 448 | 4.9 | 13 | 13,031 | 13,031 | −0.7 | 16 |
| n60w140.004 | 488 | 13,189 | 488 | 488 | 0.0 | 15 | 12,663 | 12,663 | −4.0 | 15 |
| n60w160.002 | 423 | 12,471 | 423 | 423 | 0.0 | 19 | 12,719 | 12,719 | 2.0 | 17 |
| n60w160.003 | 434 | 10,682 | 447 | 447 | 3.0 | 14 | 10,674 | 10,674 | −0.1 | 15 |

**Table 7 Comparison between our results with the best-found values for TSPTW and TRPTW instances proposed by *Gendreau et al. (1998)*, and *Ohlmann & Thomas (2007)*.**

| Instances | TSPTW | TRPTW | MFEA+RNVS | | | | | | | |
|---|---|---|---|---|---|---|---|---|---|---|
| | | | TSPTW | | | | TRPTW | | | |
| | OPT | KBS | Best.Sol | Aver.Sol | Gap | Time | Best.Sol | Aver.Sol | Gap | Time |
| n60w160.004 | 401 | 11,645 | 401 | 401 | 0.0 | 19 | 11,778 | 11,778 | 1.1 | 19 |
| n60w180.002 | 399 | 12,015 | 399 | 399 | 0.0 | 17 | 12,224 | 12,224 | 1.7 | 21 |
| n60w180.003 | 445 | 12,214 | 445 | 445 | 0.0 | 18 | 12,214 | 12,679 | 0.0 | 21 |
| n60w180.004 | 456 | 11,101 | 456 | 456 | 0.0 | 19 | 11,245 | 11,245 | 1.3 | 18 |
| n60w200.002 | 414 | 11,748 | 414 | 414 | 0.0 | 20 | 11,866 | 11,866 | 1.0 | 19 |
| n60w200.003 | 455 | 10,697 | 460 | 460 | 1.1 | 19 | 10,697 | 10,697 | 0.0 | 18 |
| n60w200.004 | 431 | 11,441 | 431 | 431 | 0.0 | 16 | 11,441 | 11,441 | 0.0 | 17 |
| n80w120.002 | 577 | 18,181 | 577 | 577 | 0.0 | 17 | 18,383 | 18,383 | 1.1 | 21 |
| n80w120.003 | 540 | 17,878 | 548 | 548 | 1.5 | 19 | 17,937 | 17,937 | 0.3 | 21 |
| n80w120.004 | 501 | 17,318 | 501 | 501 | 0.0 | 18 | 17,578 | 17,578 | 1.5 | 18 |
| n80w140.002 | 472 | 17,815 | 472 | 472 | 0.0 | 19 | 17,815 | 17,815 | 0.0 | 21 |
| n80w140.003 | 580 | 17,315 | 580 | 580 | 0.0 | 20 | 17,358 | 17,358 | 0.2 | 21 |
| n80w140.004 | 424 | 18,936 | 424 | 424 | 0.0 | 19 | 18,936 | 18,936 | 0.0 | 18 |
| n80w160.002 | 553 | 17,091 | 553 | 553 | 0.0 | 27 | 17,200 | 17,200 | 0.6 | 18 |
| n80w160.003 | 521 | 16,606 | 521 | 521 | 0.0 | 21 | 16,521 | 16,521 | −0.5 | 20 |
| n80w160.004 | 509 | 17,804 | 509 | 509 | 0.0 | 20 | 17,927 | 17,927 | 0.7 | 18 |
| n80w180.002 | 479 | 17,339 | 479 | 479 | 0.0 | 21 | 17,339 | 17,339 | 0.0 | 19 |
| n80w180.003 | 530 | 17,271 | 530 | 530 | 0.0 | 20 | 17,160 | 17,160 | −0.6 | 20 |
| n80w180.004 | 479 | 16,729 | 479 | 479 | 0.0 | 22 | 16,849 | 16,849 | 0.7 | 21 |
| n100w120.002 | 540 | 29,882 | 556 | 556 | 3.0 | 38 | 29,818 | 29,818 | 0.0 | 45 |
| n100w120.003 | 617 | 25,275 | 646 | 646 | 4.7 | 37 | 24,473 | 24,473 | 0.0 | 42 |
| n100w120.004 | 663 | 30,102 | 663 | 663 | 0.0 | 39 | 31,554 | 31,554 | 0.0 | 41 |
| n100w140.002 | 622 | 30,192 | 632 | 632 | 1.6 | 38 | 30,087 | 30,087 | 0.0 | 45 |
| n100w140.003 | 481 | 28,309 | 481 | 481 | 0.0 | 39 | 28,791 | 28,791 | 0.0 | 47 |
| n100w140.004 | 533 | 27,448 | 533 | 533 | 0.0 | 40 | 27,990 | 27,990 | 0.0 | 45 |
| aver | | | | | 0.64 | 13.6 | | | −0.44 | 14.7 |

space by exploiting the information that a current solution is reached in this space. In this experiment, the balance between exploration and exploitation is considered. We run the MFEA with or without the RVNS on the same selected instances. In Table 3, the MFEA-NLS column is the MFEA without the RVNS, while the MFEA+RVNS column is the MFEA with the RVNS. The *diff* [%] column is the difference between the MFEA+RVNS and MFEA-NLS.

In Table 3, the MFEA+RVNS obtains much better solutions than the MFEA-NLS in all cases. It indicates that the combination between the MFEA and RNVS has a good balance between exploration and exploitation. To study the ability to balance exploration and

**Table 8  Comparison between our results with the best-found values for TSPTW and TRPTW instances proposed by *Gendreau et al. (1998)*, and *Ohlmann & Thomas (2007)*.**

| Instances | TSPTW | TRPTW | | MFEA+RNVS | | | | | | |
|---|---|---|---|---|---|---|---|---|---|---|
| | | | | TSP | | | TRP | | | |
| | *OPT* | BKS | | *Best.Sol* | *Aver.Sol* | *Time* | *Best.Sol* | *Aver.Sol* | *Time* | |
| | | *Ban (2021)* | *Heilporna, Cordeaua & Laporte (2010)* | | | | | | | |
| n20w120.001 | 274 | 2,175 | 2,535 | 274 | 274 | 2 | 2,175 | 2,175 | 2 | |
| n20w140.001 | 176 | 1,846 | 1,908 | 176 | 176 | 2 | 1,826 | 1,826 | 2 | |
| n20w160.001 | 241 | 2,146 | 2,150 | 241 | 241 | 2 | 2,148 | 2,148 | 2 | |
| n20w180.001 | 253 | 2,477 | 2,037 | 253 | 253 | 2 | 2,477 | 2,477 | 2 | |
| n20w200.001 | 233 | 1,975 | 2,294 | 233 | 233 | 2 | 1,975 | ,1975 | 2 | |
| n40w120.001 | 434 | 6,800 | 7,496 | 434 | 434 | 9 | 6,800 | 6,800 | 9 | |
| n40w140.001 | 328 | 6,290 | 7,203 | 328 | 328 | 10 | 6,290 | 6,290 | 10 | |
| n40w160.001 | 349 | 6,143 | 6,657 | 349 | 349 | 11 | 6,143 | 6,143 | 12 | |
| n40w180.001 | 345 | 6,952 | 6,578 | 345 | 345 | 12 | 6,897 | 6,897 | 11 | |
| n40w200.001 | 345 | 6,169 | 6,408 | 345 | 345 | 10 | 6,113 | 6,113 | 13 | |
| n60w120.001 | 392 | 11,120 | 9,303 | 392 | 392 | 25 | 11,288 | 11,288 | 28 | |
| n60w140.001 | 426 | 10,814 | 9,131 | 426 | 426 | 26 | 10,981 | 10,981 | 27 | |
| n60w160.001 | 589 | 11,574 | 11,422 | 589 | 589 | 27 | 11,546 | 11,546 | 28 | |
| n60w180.001 | 436 | 11,363 | 9,689 | 436 | 436 | 24 | 11,646 | 11,646 | 25 | |
| n60w200.001 | 423 | 10,128 | 10,315 | 423 | 423 | 25 | 9,939 | 9,939 | 27 | |
| n80w120.001 | 509 | 11,122 | 11,156 | 512 | 509 | 41 | 16,693 | 16,693 | 45 | |
| n80w140.001 | 530 | 14,131 | 14,131 | 530 | 530 | 42 | 14,131 | 14,131 | 47 | |
| n80w180.001 | 605 | 11,222 | 11,222 | 605 | 605 | 41 | 11,222 | 11,222 | 42 | |

**Table 9  The average results for TSPTW, TRPTW instances.**

| Instances | TSPTW | | | TRPTW | |
|---|---|---|---|---|---|
| | $\overline{gap}$ | *Time* | $\overline{gap}$ | *Time* | |
| TSPTW | 0.49 | 26.24 | −0.11 | 26.5 | |
| TSPTW | 0.64 | 13.6 | −0.44 | 14.7 | |
| aver | 0.56 | 19.9 | −0.28 | 20.6 | |

exploitation of the search space, we implement an experimental study on the distribution of locally optimal solutions. We choose two instances (n20w100.002 and n40w100.002) to perform one execution of our algorithm and record the distinct local optima encountered in some generations. We then plot the normalized tour's cost *vs* its average metric distance to all other local minima (the distance metric and its average is defined in "Selection operator"). The results are illustrated in Figs. 4–7. The black "x" points indicate the result

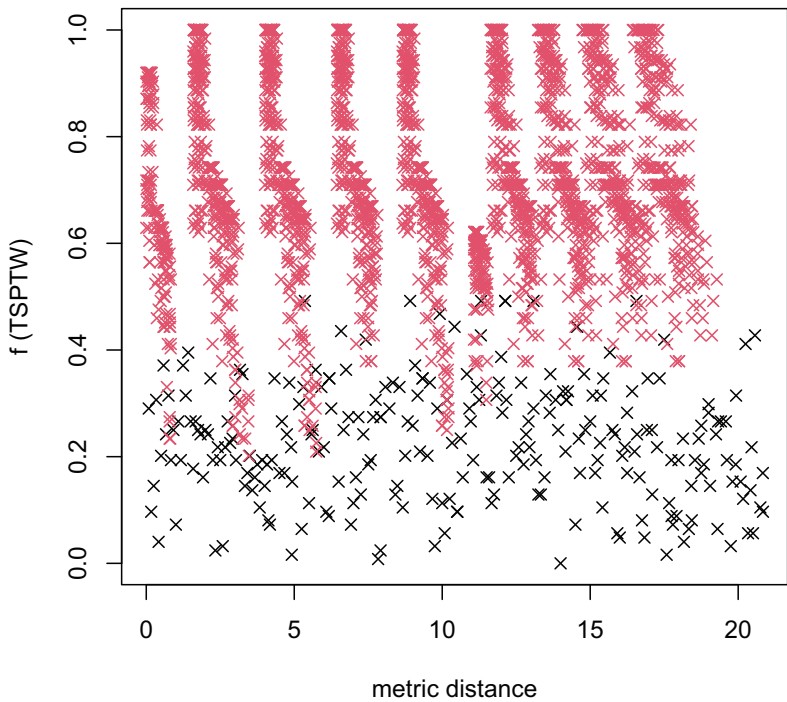

**Figure 4 The average distance to the other local optima in n20w100.002 instance for the TSPTW.**

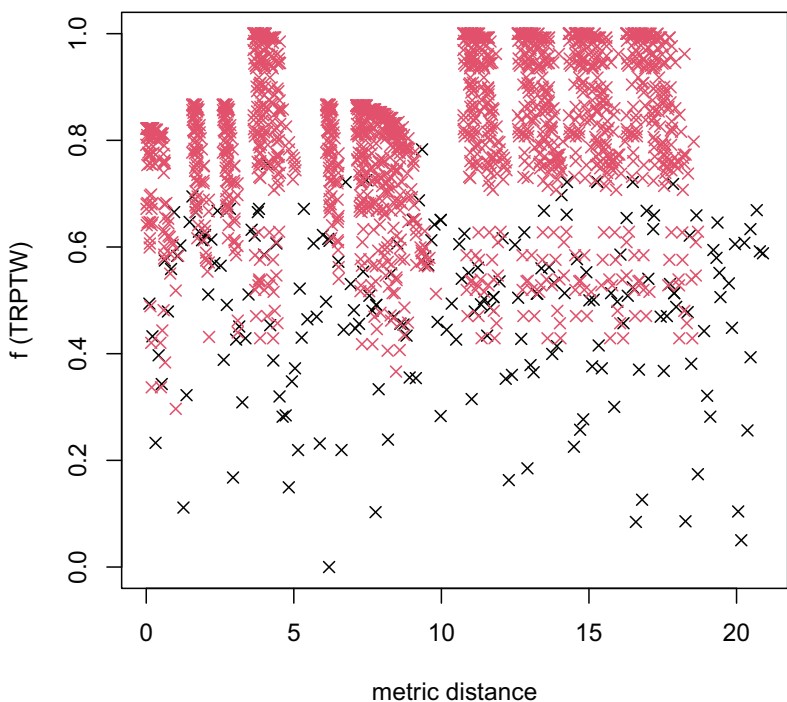

**Figure 5 The average distance to the other local optima in n20w100.002 instance for the TRPTW.**

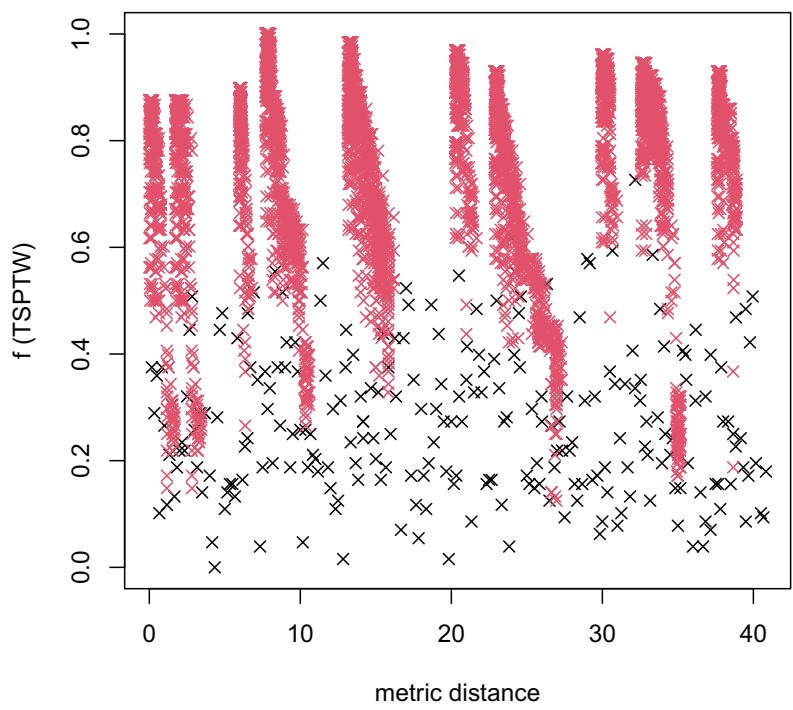

**Figure 6 The average distance to the other local optima in n40w100.002 instance for the TSPTW.**

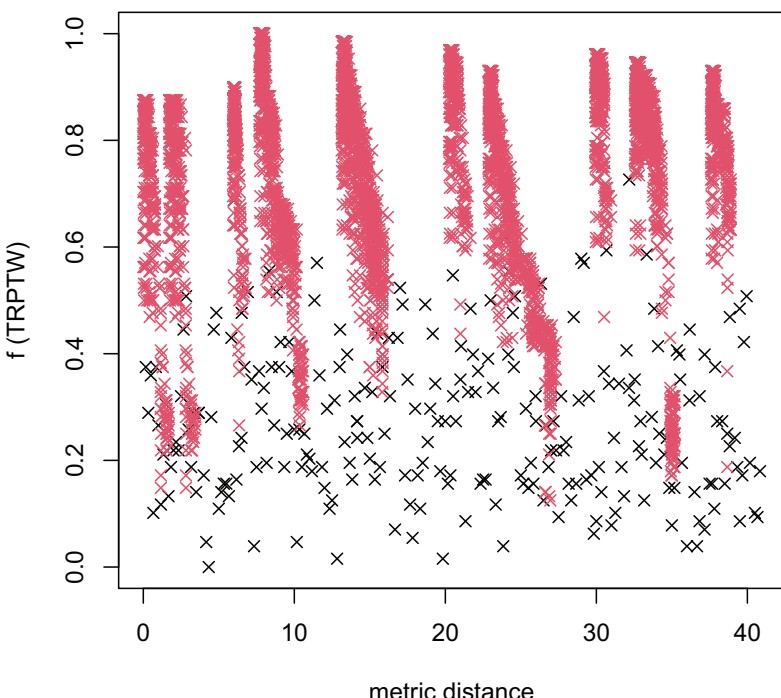

**Figure 7 The average distance to the other local optima in n40w100.002 instance for the TRPTW.**

of the MFEA, while the red "x" points show the results of the RNVS. The normalized cost can be used as follows:

$$\overline{f_j} = \frac{(f_j - f_j^{min})}{(f_j^{max} - f_j^{min})}, \qquad (4)$$

where $j = 1, 2$ is the $j$-task and $f_j^{min}, f_j^{max}$ are the minimum and maximum cost values for all runs, respectively.

Figures 4–7 show that the black "x" points are spread quite widely, which implies that the proposed algorithm has the power to search over a wide region of the solution space. It is the capacity for exploration. On the other hand, the red points describe the search tends to exploit the good solution space explored by the MFEA. It is the capacity for exploitation. As a result, the algorithm maintains the right balance between exploration and exploitation.

## Comparisons with the TSPTW and TRPTW algorithms

In Table 9, the average difference with the optimal solution for the TSPTW is 0.56%, even for instances with up to 200 vertices. It shows that our solutions are near-optimal for the TSPTW. In addition, the proposed algorithm reaches the optimal solutions for the instances with up to 80 vertices for the TSPTW. In Table 8, for the TRPTW, our MFEA +RVNS is better than the previous algorithms such as *Ban & Nghia (2017)*, *Ban (2021)* and *Heilporna, Cordeaua & Laporte (2010)* in the literature when the average *gap* is −0.28% (note that: Ban et al.'s and Heilporna et al.'s (2010) algorithms is developed to solve the TRPTW only). The obtained results are impressive since it can be observed that the proposed algorithm finds not only near-optimal solutions but also the new best-known ones for two problems simultaneously. It also indicates the efficiency of positive transferrable knowledge control techniques between optimization tasks in improving the solution quality.

It is impossible to expect that the MFEA+RVNS always outperform in comparison with the state-of-the-art metaheuristic algorithms for the TSPTW and TRPTW in all cases because their algorithms are designed to solve each problem independently. Table 10 shows that the efficient algorithms for the TSPTW may not be effective for the TRPTW and vice versa. On average, the optimal solution for the TSPTW with the TRPTW objective cost is about 9.7% worse than the optimal one for the TRPTW. Similarly, the known best solution for the TRPTW using the TSPTW objective function is 20.6% worse than the optimal solution for the TSPTW. We conclude that if the proposed MFEA simultaneously reaches good solutions that are close to the optimal solutions for both problems and even better than the state-of-the-art algorithms in many cases, we can say that the proposed MFEA+RVNS for multitasking is beneficial.

## Comparison with the previous MFEA algorithms

We adopt the proposed algorithm in the experiment to solve the TSP and TRP problems. Otherwise, we also use three algorithms (*Ban & Pham, 2022*; *Osaba et al., 2020*; *Yuan et al., 2016*) to solve the TSPTW and TRPTW.

**Table 10 The difference between the optimal TSPTW using TRPTW objective function and *vice versa*.**

| Instances | TRPTW | | | TSPTW | | |
|---|---|---|---|---|---|---|
| | TRPTW (OPT_TSPTW) | KBS | diff[%] | TSPTW (BKS_TRPTW) | KBS | diff[%] |
| n20w120.002 | 2,592 | 2,193 | 15.4 | 256 | 218 | 14.8 |
| n20w140.002 | 2,519 | 2,330 | 7.5 | 311 | 272 | 12.5 |
| n20w160.002 | 2,043 | 1,830 | 10.4 | 249 | 201 | 19.3 |
| n40w120.002 | 6,718 | 6,265 | 6.7 | 552 | 446 | 19.2 |
| n40w140.002 | 5,865 | 5,746 | 2.0 | 449 | 383 | 14.7 |
| n40w160.002 | 7,519 | 6,351 | 15.5 | 456 | 338 | 25.9 |
| n60w120.002 | 13,896 | 1,2517 | 9.9 | 581 | 444 | 23.6 |
| n60w140.002 | 12,898 | 11,795 | 8.6 | 616 | 464 | 24.7 |
| n60w160.002 | 14,091 | 12,489 | 11.4 | 616 | 428 | 30.5 |
| aver | | | 9.7 | | | 20.6 |

**Table 11 Comparison between our results with the others for TSP and TRP (*Yuan et al., 2016*).**

| Instances | OPT | | YA (*Yuan et al., 2016*) | | OA (*Tsitsiklis, 1992*) | | MFEA+RNVS | | | | | | | |
|---|---|---|---|---|---|---|---|---|---|---|---|---|---|---|
| | TSP | TRP | TSP | TRP | TSP | TRP | TSP | | | | TRP | | | |
| | | | best.sol | best.sol | best.sol | best.sol | best.sol | aver.sol | Gap | Time | best.sol | aver.sol | Gap | Time |
| eil51 | 426* | 10,178* | 446 | 10,834 | 450 | 10,834 | 426 | 426 | 0.00 | 23 | 10,178 | 10,178 | 0.00 | 22 |
| berlin52 | 7,542* | 143,721* | 7,922 | 152,886 | 8,276 | 152,886 | 7,542 | 7,542 | 0.00 | 22 | 143,721 | 143,721 | 0.00 | 21 |
| st70 | 675* | 20,557* | 713 | 22,283 | 772 | 22,799 | 680 | 680 | 0.01 | 41 | 22,283 | 22,283 | 8.40 | 39 |
| eil76 | 538* | 17,976* | 560 | 18,777 | 589 | 18,008 | 559 | 559 | 0.04 | 43 | 18,008 | 18,008 | 0.18 | 40 |
| pr76 | 108,159* | 3,455,242* | 113,017 | 3,493,048 | 117,287 | 3,493,048 | 108,159 | 108,159 | 0.00 | 47 | 3,455,242 | 3,455,242 | 0.00 | 45 |
| pr107 | 44,303* | 2,026,626* | 45,737 | 2,135,492 | 46,338 | 2,135,492 | 45,187 | 45,187 | 0.02 | 71 | 2,052,224 | 2,052,224 | 1.26 | 72 |
| rat99 | 1,211* | 58,288* | 1,316 | 60,134 | 1,369 | 60,134 | 1,280 | 1,280 | 0.06 | 66 | 58,971 | 58,971 | 1.17 | 65 |
| kroA100 | 21,282* | 983,128* | 22,233 | 1,043,868 | 22,233 | 1,043,868 | 21,878 | 21,878 | 0.03 | 63 | 1,009,986 | 1,009,986 | 2.73 | 63 |
| kroB100 | 22,141* | 986,008* | 23,144 | 1,118,869 | 24,337 | 1,118,869 | 23,039 | 23,039 | 0.04 | 64 | 1,003,107 | 1,003,107 | 1.73 | 63 |
| kroC100 | 20,749* | 961,324* | 22,395 | 1,026,908 | 23,251 | 1,026,908 | 21,541 | 21,541 | 0.04 | 68 | 1,007,154 | 1,007,154 | 4.77 | 66 |
| kroD100 | 21,294* | 976,965* | 22,467 | 1,069,309 | 23,833 | 1,069,309 | 22,430 | 22,430 | 0.05 | 70 | 1,019,821 | 1,019,821 | 4.39 | 72 |
| kroE100 | 22,068* | 971,266* | 22,960 | 1,056,228 | 23,622 | 1,056,228 | 22,964 | 22,964 | 0.04 | 60 | 1,034,760 | 1,034,760 | 6.54 | 64 |
| rd100 | 7,910* | 340,047* | 8,381 | 3,80,310 | 8,778 | 365,805 | 8,333 | 8,333 | 0.05 | 63 | 354,762 | 354,762 | 4.33 | 64 |
| eil101 | 629* | 27,519* | 681 | 28,398 | 695 | 28,398 | 662 | 662 | 0.05 | 62 | 27,741 | 27,741 | 0.81 | 61 |
| aver | | | | | | | | | 0.03 | | | | 2.59 | |

**Note:**
* Indicates the optimal values.

Tables 11–13 compare our results to those of three algorithms (*Ban & Pham, 2022*; *Osaba et al., 2020*; *Yuan et al., 2016*) for some instances in both the TSP and TRP problems. The results show that the proposed algorithm obtains better solutions than the others in all cases. The difference between our average result and the optimal value is below 2.59%. It shows that our solution is the very near-optimal one. In addition, our algorithm

**Table 12 Comparison between our results with the others for TSP and TRP in TRP-50-x.**

| Instances | OPT | | YA | | OA | | MFEA+RNVS | |
|---|---|---|---|---|---|---|---|---|
| | TSP best.sol | TRP best.sol | TSP best.sol | TRP best.sol | TSP best.sol | TRP best.sol | TSP best.sol | TRP best.sol |
| TRP-50-1 | 602 | 12,198 | 641 | 13,253 | 634 | 13,281 | 610 | 12,330 |
| TRP-50-2 | 549 | 11,621 | 583 | 12,958 | 560 | 12,543 | 560 | 11,710 |
| TRP-50-3 | 584 | 12,139 | 596 | 13,482 | 596 | 13,127 | 592 | 12,312 |
| TRP-50-4 | 603 | 13,071 | 666 | 14,131 | 613 | 15,477 | 610 | 13,575 |
| TRP-50-5 | 557 | 12,126 | 579 | 13,377 | 578 | 14,449 | 557 | 12,657 |
| TRP-50-6 | 577 | 12,684 | 602 | 13,807 | 600 | 13,601 | 588 | 13,070 |
| TRP-50-7 | 534 | 11,176 | 563 | 11,984 | 555 | 12,825 | 547 | 11,793 |
| TRP-50-8 | 569 | 12,910 | 629 | 14,043 | 609 | 13,198 | 572 | 13,198 |
| TRP-50-9 | 575 | 13,149 | 631 | 14,687 | 597 | 13,459 | 576 | 13,459 |
| TRP-50-10 | 583 | 12,892 | 604 | 14,104 | 602 | 13,638 | 590 | 13,267 |
| TRP-50-11 | 578 | 12,103 | 607 | 13,878 | 585 | 12,124 | 585 | 12,124 |
| TRP-50-12 | 500 | 10,633 | 521 | 11,985 | 508 | 11,777 | 604 | 11,305 |
| TRP-50-13 | 579 | 12,115 | 615 | 13,885 | 601 | 13,689 | 587 | 12,559 |
| TRP-50-14 | 563 | 13,117 | 612 | 14,276 | 606 | 14,049 | 571 | 13,431 |
| TRP-50-15 | 526 | 11,986 | 526 | 12,546 | 526 | 12,429 | 526 | 12,429 |
| TRP-50-16 | 551 | 12,138 | 577 | 13,211 | 564 | 12,635 | 551 | 12,417 |
| TRP-50-17 | 550 | 12,176 | 601 | 13,622 | 585 | 13,342 | 564 | 12,475 |
| TRP-50-18 | 603 | 13,357 | 629 | 14,750 | 625 | 14,108 | 603 | 13,683 |
| TRP-50-19 | 529 | 11,430 | 595 | 12,609 | 594 | 12,899 | 539 | 11,659 |
| TRP-50-20 | 539 | 11,935 | 585 | 13,603 | 575 | 12,458 | 539 | 12,107 |

reaches the optimal solution for the instance with 76 vertices. Obviously, the proposed algorithm applies well in the case of the TSP and TRP.

Statistical tests are used to check whether the difference between the proposed algorithm and the remaining ones is significant or not. A non-parametric test (Friedman and Quad test) is carried out in the group of the algorithms to check if a significant difference between them is found. The output of the Friedman test in Table 14 illustrates the rankings achieved by the Friedman and Quade tests. The results in this table strongly indicate considerable differences between the algorithms. Because the YA and OA have the largest ranking, the MFEA+RNVS is selected as the control algorithms. After that, we compare the control algorithms with YA and OA by statistical methods. Table 15 shows the possible hypotheses of comparison between the control algorithm and other algorithms. The statistical result shows that the MFEA+RNVS outperforms YA and OA with a level of significance $\alpha = 0.05$.

Figures 8–13 describe the normalized tour's cost *vs* its average metric distance to all other local minima of two algorithms (YA and OA). The black and red points are still the results of the MFEA and RVNS, respectively. Their algorithm explores widely in the solution space, implying a good capacity for exploration. However, exploitation capacity is

**Table 13 Comparison between our results with the others for TSP and TRP in TRP-100-x.**

| Instances | OPT | | YA | | OA | | MFEA+RNVS | |
|---|---|---|---|---|---|---|---|---|
| | TSP best.sol | TRP best.sol | TSP best.sol | TRP best.sol | TSP best.sol | TRP best.sol | TSP best.sol | TRP best.sol |
| TRP-100-1 | 762 | 32,779 | 830 | 36,012 | 806 | 36,869 | 791 | 35,785 |
| TRP-100-2 | 771 | 33,435 | 800 | 39,019 | 817 | 37,297 | 782 | 35,546 |
| TRP-100-3 | 746 | 32,390 | 865 | 38,998 | 849 | 34,324 | 767 | 34,324 |
| TRP-100-4 | 776 | 34,733 | 929 | 41,705 | 897 | 38,733 | 810 | 37,348 |
| TRP-100-5 | 749 | 32,598 | 793 | 40,063 | 899 | 37,191 | 774 | 34,957 |
| TRP-100-6 | 807 | 34,159 | 905 | 40,249 | 886 | 40,588 | 854 | 36,689 |
| TRP-100-7 | 767 | 33,375 | 780 | 38,794 | 849 | 39,430 | 780 | 35,330 |
| TRP-100-8 | 744 | 31,780 | 824 | 38,155 | 845 | 35,581 | 763 | 34,342 |
| TRP-100-9 | 786 | 34,167 | 863 | 39,189 | 858 | 41,103 | 809 | 35,990 |
| TRP-100-10 | 751 | 31,605 | 878 | 36,191 | 831 | 37,958 | 788 | 33,737 |
| TRP-100-11 | 776 | 34,188 | 831 | 39,750 | 876 | 41,153 | 814 | 36,988 |
| TRP-100-12 | 797 | 32,146 | 855 | 39,422 | 855 | 40,081 | 823 | 34,103 |
| TRP-100-13 | 753 | 32,604 | 772 | 37,004 | 772 | 40,172 | 771 | 35,011 |
| TRP-100-14 | 770 | 32,433 | 810 | 40,432 | 810 | 36,134 | 800 | 34,576 |
| TRP-100-15 | 776 | 32,574 | 953 | 38,369 | 878 | 38,450 | 810 | 35,653 |
| TRP-100-16 | 775 | 33,566 | 838 | 40,759 | 835 | 38,549 | 808 | 36,188 |
| TRP-100-17 | 805 | 34,198 | 939 | 39,582 | 881 | 42,155 | 838 | 36,969 |
| TRP-100-18 | 785 | 31,929 | 876 | 38,906 | 836 | 37,856 | 814 | 34,154 |
| TRP-100-19 | 780 | 33,463 | 899 | 39,865 | 881 | 40,379 | 797 | 35,669 |
| TRP-100-20 | 775 | 33,632 | 816 | 41,133 | 905 | 40,619 | 808 | 35,532 |

**Table 14 Average rankings of the algorithms.**

| Algorithm | TSP | | TSP | |
|---|---|---|---|---|
| | Friedman | Quade | Friedman | Quade |
| YA | 2.73 | 2.70 | 2.59 | 2.52 |
| OA | 2.15 | 2.18 | 2.33 | 2.43 |
| MFEA+RNVS | 1.11 | 1.11 | 1.06 | 1.04 |

**Table 15 The $z$-values and $p$-values of the Friedman procedures (MFEA+RNVS is the control algorithm) in both the TSP and TRP.**

| $i$ | Algorithm | TSP | | | | | TRP | | | | |
|---|---|---|---|---|---|---|---|---|---|---|---|
| | | $z$ | $p$ | Holm | Holland | Rom | $z$ | $p$ | Holm | Holland | Rom |
| 1 | YA | 7.26 | 3.66E−13 | 0.025 | 0.025 | 0.025 | 6.87 | 6.15E−12 | 0.025 | 0.025 | 0.025 |
| 2 | OA | 4.63 | 3.48E−06 | 0.05 | 0.05 | 0.05 | 5.70 | 1.18E−08 | 0.05 | 0.05 | 0.05 |

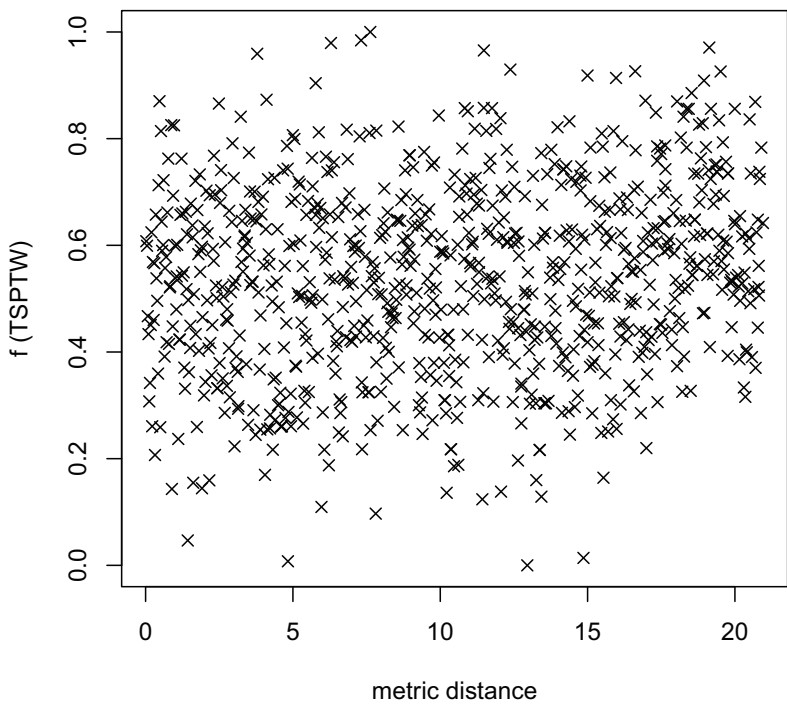

**Figure 8 The average distance to the other local optima in n20w100.002 instance for the TSP (YA algorithm).**

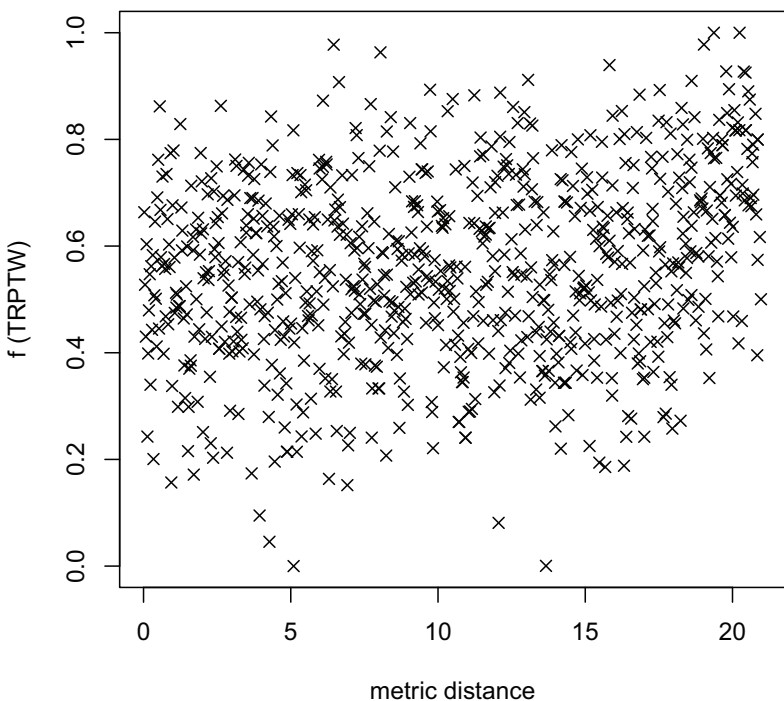

**Figure 9 The average distance to the other local optima in n20w100.002 instance for the TRP (YA algorithm).**

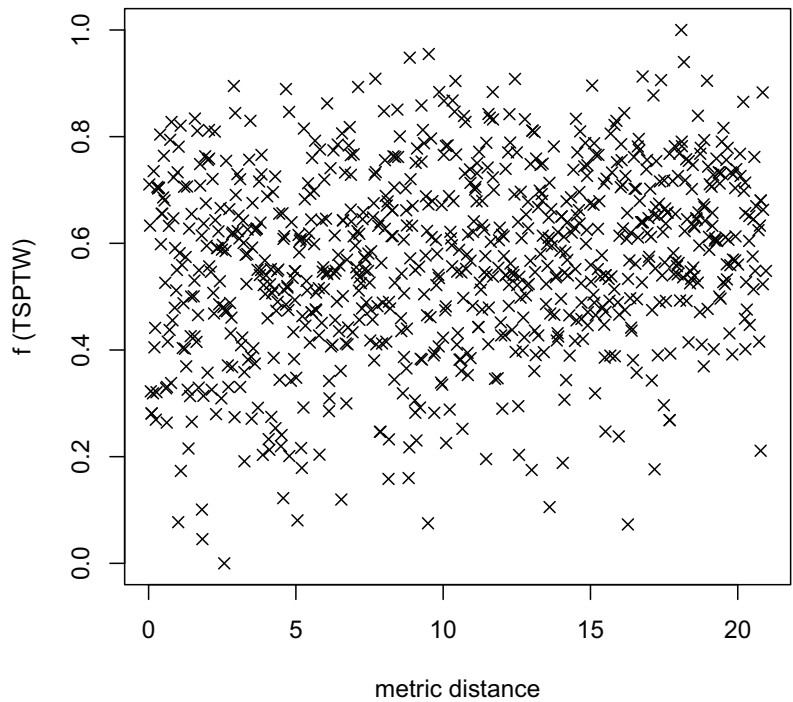

**Figure 10** **The average distance to the other local optima in n20w100.002 instance for the TSP (OA algorithm).**

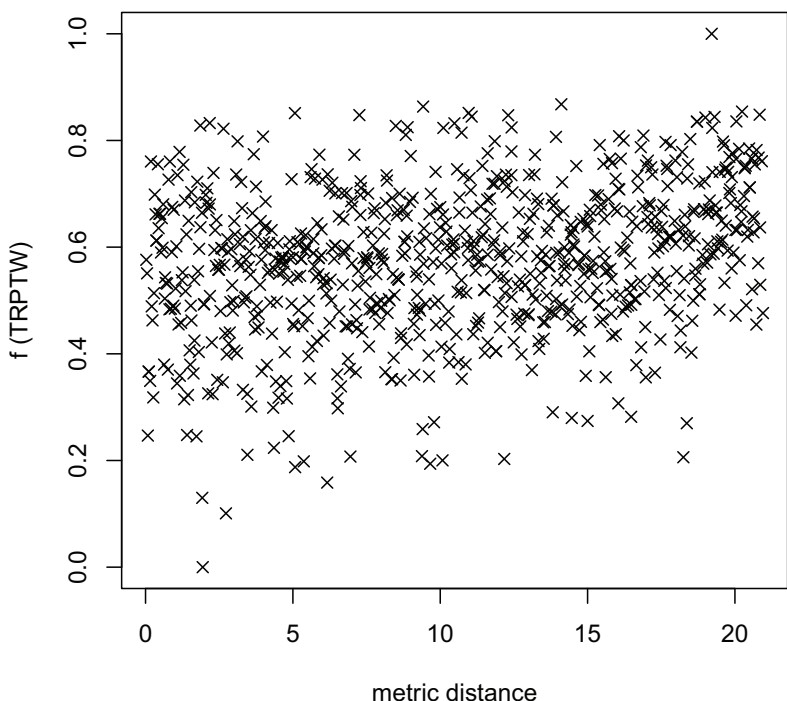

**Figure 11** **The average distance to the other local optima in n20w100.002 instance for the TRP (OA algorithm).**

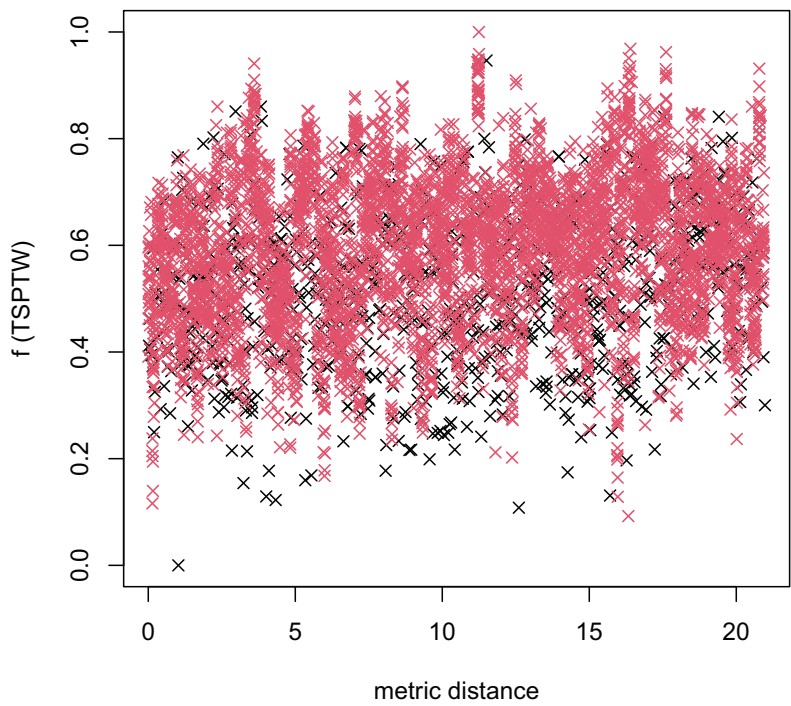

**Figure 12** The average distance to the other local optima in n20w100.002 instance for the TSP (the proposed algorithm).

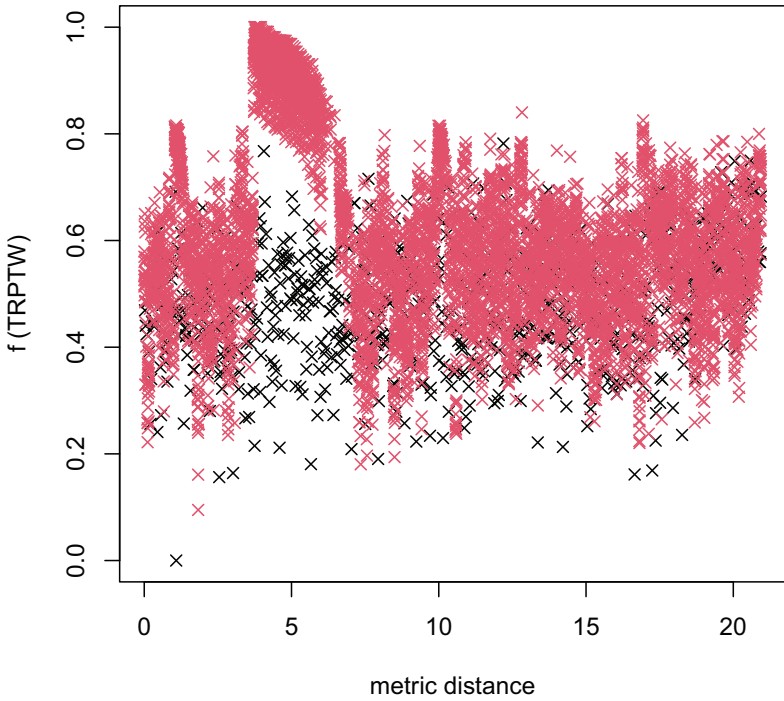

**Figure 13** The average distance to the other local optima in n20w100.002 instance for the TRP (the proposed algorithm).

**Table 16 Comparison between our results with the others (*Ban & Pham, 2022*; *Osaba et al., 2020*; *Yuan et al., 2016*) for TSPTW and TRPTW.**

| Instances | YA (*Xu et al., 2021*) | | OA (*Tsitsiklis, 1992*) | | BP (*Ban & Pham, 2022*) | | MFEA+RNVS | |
|---|---|---|---|---|---|---|---|---|
| | TSPTW | TRPTW | TSPTW | TRPTW | TSPTW | TRPTW | TSPTW | TRPTW |
| n40w40.002 | – | – | – | – | – | – | 461 | 7,104 |
| n40w60.002 | – | – | – | – | – | – | 470 | 7,247 |
| n40w80.002 | – | – | – | – | – | – | 431 | 7,123 |
| n40w100.002 | – | – | – | – | – | – | 364 | 6,693 |
| n60w20.002 | – | – | – | – | – | – | 605 | 13,996 |
| n60w120.002 | – | – | – | – | – | – | 427 | 12,525 |
| n60w140.002 | – | – | – | – | – | – | 464 | 11,810 |
| n60w160.002 | – | – | – | – | – | – | 423 | 12,719 |
| n80w120.002 | – | – | – | – | – | – | 577 | 18,383 |
| n80w140.002 | – | – | – | – | – | – | 472 | 18,208 |
| n80w160.002 | – | – | – | – | – | – | 553 | 17,200 |
| n100w120.002 | – | – | – | – | – | – | 556 | 29,818 |
| n100w140.002 | – | – | – | – | – | – | 632 | 30,087 |
| n100w120.003 | – | – | – | – | – | – | 646 | 24,473 |
| n100w140.003 | – | – | – | – | – | – | 481 | 28,791 |
| n100w140.004 | – | – | – | – | – | – | 533 | 27,990 |

**Table 17 Comparison computational effort between single-tasking and multitasking.**

| Type | TSPTW gap | TRPTW gap |
|---|---|---|
| Single-tasking (100 generations) | 0.59 | 0.08 |
| Multi-tasking (100 generations) | 0.56 | −0.28 |

not good enough. Figures 12 and 13 show that the RVNS exploits good solution spaces explored by the MFEA much better. It is understandable when the proposed algorithm reaches better results than theirs.

Moreover, Table 16 compares our results to those of three algorithms (*Ban & Pham, 2022*; *Osaba et al., 2020*; *Yuan et al., 2016*) in both the TSPTW and TRPTW problems. The results show that three algorithms (*Ban & Pham, 2022*; *Osaba et al., 2020*; *Yuan et al., 2016*) cannot find feasible solutions while the proposed algorithm reaches feasible ones in all cases. It is understandable because these algorithms drive the search for solution spaces that maybe not contain feasible solutions. Otherwise, the proposed algorithm guides the search process to feasible solution spaces. It is an important contribution because finding a feasible solution for the TSPTW, and TRPTW is even NP-hard (*Ban, 2021*).

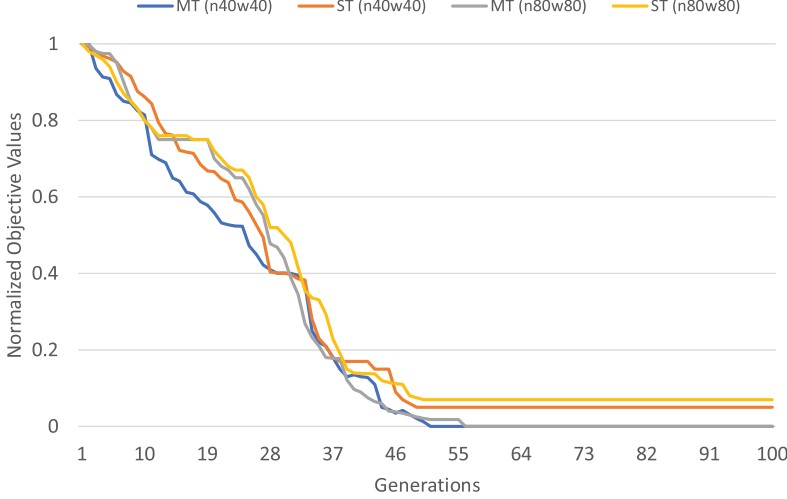

**Figure 14** Comparing convergence trends of $f_1$ in multi-tasking and single-tasking for the TSPTW in n40w40 and n80w80 instances.

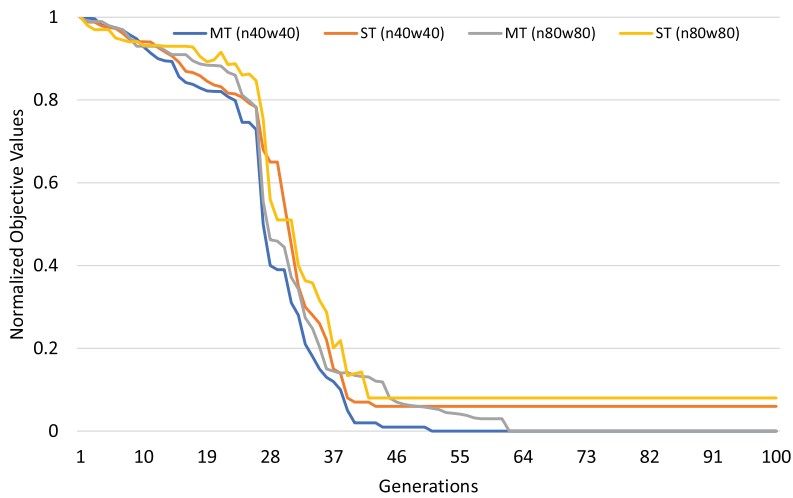

**Figure 15** Comparing convergence trends of $f_2$ in multi-tasking and single-tasking for the TRPTW in n40w40 and n80w80 instances.

## Convergence trend

The normalized objective cost (see formulation 4) can be used to analyze the convergence trends of our MFEA+RVNS algorithm. The convergence trend of the two strategies is described in Figs. 4 and 5 for n40w40 and n80w80 instances. The x-axis describes the number of generations, while the y-axis illustrates the normalized objective value. The less and less the normalized objective value is, the better and better the algorithm is. In Figs. 14 and 15, Single-tasking (ST) converges better than multitasking (MT) in the whole evolutionary process while avoiding premature convergence to sub-optimal solutions by exchanging knowledge among tasks. That means, in general, multitasking converges to a better objective value.

When multitasking is run with the same number of generations as single-tasking, on average, it only consumes $\frac{1}{K}$ computational effort for each task ($K$ is the number of tasks). Therefore, we consider the worst-case situation when the number of generations for multitasking is $K$ times the one for single-tasking. If multitasking obtains better solutions than single-tasking in this case, we can say that multitasking saves computational efforts. The experimental results are described in Table 17. In Table 17, the first row shows the average gap of single-task for the TSPTW and TRPTW, while the second shows the average gap of multitasking with 100 generations. The result shows that multitasking consumes only 1/2 computational efforts to obtain better solutions than single-tasking.

In short, the efficiency of multitasking is better in comparison with single-tasking because of the process of transferring knowledge during multitasking. It is an impressive advantage of the evolutionary multitasking paradigm.

## CONCULSIONS AND FUTURE WORK

In this article, our contribution is threefold. Firstly, we propose a new selection operator that balances skill-factor and population diversity. The skill-factor effectively transfers elite genes between tasks, while diversity in the population is important when it meets a bottleneck against the information transfer. Secondly, multiple crossover schemes are applied in the proposed MFEA+RVNS. They help the algorithm have good enough diversity. In addition, two types of crossover (intra- and inter-) are used. It opens up the chance for knowledge transfer through crossover-based exchange between tasks. Lastly, the combination between the MFEA and the RVNS is to have good transferrable knowledge between tasks from the MFEA and the ability to exploit good solution spaces from the RVNS. However, focusing only on reducing cost function maybe lead the search to infeasible solution spaces like the algorithm (*Ban & Pham, 2022*). Therefore, the repair technique is incorporated into the proposed algorithm to balance finding feasible solution spaces and reducing cost function.

Extensive experiments on the benchmark dataset indicate that the proposed algorithm simultaneously obtains good solutions for both problems. In addition, it obtains better solutions than the other MFEA algorithms in many cases. More interestingly, it finds the new best-known solutions compared to the state-of-the-art metaheuristics only for the TRPTW in many cases.

In future work, we will study how to apply multiple population ideas for multitasking. Many researchers is interested in the approach (*Chen et al., 2020*; *Li, Zhang & Gao, 2018*; *Wei & Zhong, 2020*; *Wei et al., 2022*; *Xu et al., 2021*). The approach's advantages include: (1) each population evolves with different genetic operators, and each individual can be represented differently; (2) individuals migrate between populations. The approach maintains diversity and improves convergence trends.

### Funding
This research was supported by the Hanoi University of Science and Technology under grant number T2021-PC-021. The funders had no role in study design, data collection and analysis, decision to publish, or preparation of the manuscript.

### Grant Disclosures
The following grant information was disclosed by the authors:
Hanoi University of Science and Technology: T2021-PC-021.

### Competing Interests
The authors declare that they have no competing interests.

### Author Contributions
- Ha-Bang Ban conceived and designed the experiments, performed the experiments, analyzed the data, performed the computation work, prepared figures and/or tables, authored or reviewed drafts of the article, and approved the final draft.
- Dang-Hai Pham conceived and designed the experiments, analyzed the data, prepared figures and/or tables, and approved the final draft.

### Data Availability
The data and code are available at Zenodo: Ha-Bang Ha, & Dang-Hai Pham. (2022). Dataset and code for the variants of the traveling salesman problem with time windows using multifactorial evolutionary algorithm [Data set]. In PeerJ Computer Science. Zenodo. https://doi.org/10.5281/zenodo.7331323.

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
