# Peer review of "Solving optimization problems simultaneously: the variants of the traveling salesman problem with time windows using multifactorial evolutionary algorithm"

_PeerJ Computer Science, doi:10.7717/peerj-cs.1192_

## Round 0.1 · original submission · Major Revisions

There are some concerns about the writing and presentation of the paper. Please address all comments raised by the reviewers and prepare a detailed response letter. Thanks.

Reviewer 1 ·

Basic reporting

1. The structure of this paper is very poor. I suggest the sections 2.1 and 2.2 should be moved to section 1 and the description of two problems should be moved to section 2. The section should be moved to section 2 and be concise.
2. Too many references have been provided for only one thing, such as lines 30-31, 37-40, etc. I think it is unsuitable, and I suggest only the main references should be given.
3. The section 2.1 is too simple, and the previous studies should be analyzed more comprehensively.
4. The section 6 is not the discussion study and should be concise.

Experimental design

1. The section 4 is very confused, and I can not know what is the authors works.
2. The results in section 5 should be analyzed more deeply.

Validity of the findings

no comment

Additional comments

1. The Abstract is very poor and should be rewritten. The object, method and results should be provided in the Abstract.
2. Line 86, what is a dMFEA-II framework?
3. The format of table 8 should be improved.
4. The format of the references should be uniform.

Reviewer 2 ·

Basic reporting

In general, authors complied with reviewers' requests in the first round. However, I suggest other improvements in the paper:

1) The results can be better described:
a) For example, expand the explanation of the results in Tables 2 to 8. The authors include many tables, but little description is given of the data in these structures.
b) Figures 3 and 4: should also be better explained.
2) Include suggestions for future work.
3) Also include other more recent references: last three years.
4) The link https://homepages.dcc.ufmg.br/rfsilva/tsptw/ is incorrect. Change by https://homepages.dcc.ufmg.br/~rfsilva/tsptw/

Experimental design

Ok.

Validity of the findings

Ok.

Reviewer 3 ·

Basic reporting

In " Solving Optimization Problems Simultaneously: The variants of the Traveling Salesman Problem with Time Windows using Multifactorial Evolutionary Algorithm”, the authors performed a combination of Evolutionary Algorithms to tackle Traveling Salesman Problem and some related problem. The work deals with an important topic and it might be well received by the scientific community. However, there some important points that need to be addressed before a positive recommendation.

1. More details should be given to initialization process. For instance, Algorithm 1 there is no clue how ‘start vertex’ are calculated.

2. The results on time computation should be better described. How much time has been saved in average?

3. Information about hardware and software should be provide to allow reproducibility.

4. How the parameters of the algorithm have been determined (Table 1)?

5. Minor problems

- Figures are poor described, and legend should be self-contained. For instance, Figure 1 should have more details to clarify the difference.
- References are not respecting margins.
- There are few references from 2021 and 2020. I wander if the authors could clarify how the state-of-the-art was made and if there are or are not more current references.

In case the manuscript passes to 2nd round, I will be happy to review the manuscript again.

Experimental design

See my main comments.

Validity of the findings

See my main comments.

Additional comments

See my main comments.

---

## Round 0.2 · Major Revisions

A major revision is still needed to address the comments of one reviewer. Please provide a detailed response letter. Thanks.

Reviewer 1 ·

Basic reporting

no comment

Experimental design

no comment

Validity of the findings

no comment

Additional comments

I have read the revision again, and found that there are still many serious defects. The main comments are as follows,
1. The revision is not marked in the paper, and I can not find them easily. I think the revised parts should be shown clearly.
2. Too many references have been provided for only one thing, such as lines 32-34, 124-125, etc. I think it is unsuitable, and I suggest only the main references should be given.
3. The section 1 is too simple, and the previous studies should be analyzed more comprehensively.
4. The sections 2 and 3 should be combined.
5. Line 65, what is a dMFEA-II framework?
6. There are two flow charts in the figure 1. Thus, two subtitles should be provided.
7. Line 151, "The pseudo-code of the basic MFEA is described in Figure 1". I find the flow chart in figure 1 and not pseudo-code.
8. Lines 290-291, I think it should be one step of the algorithm.
9. The results in section 5 should be analyzed more deeply.
10. The section 5.6 is not the discussion study and should be combined with other sections.
11. The section 6 is very bad and should be rewritten. The results and new findings should be provided clearly.
12. The format of the references should be uniform.

Reviewer 2 ·

Basic reporting

The authors carried out all my suggestions in this new version.

Experimental design

Ok.

Validity of the findings

Ok.

---

## Round 0.3 · Major Revisions

The reviewer recommended major revisions in this round. Please address the comments. I will make a decision regarding publication or rejection in the next round.

Reviewer 1 ·

Basic reporting

no

Experimental design

no

Validity of the findings

no

Additional comments

1. The section 1 is still simple, and the previous studies should be analyzed more deeply and comprehensively.
2. Line 75, I can not find the explanation on dMFEA in main text.
3. I want to point out that the figure 1 is the flow chart and not pseudo-code. Please provide the pseudo-code of MFEA.
4. The results in section 4 should be analyzed more deeply.
5. The format of the references should be uniform.

---

## Round 0.4 · accepted · Accept

The reviewers' comments have been addressed. The paper can be accepted now.

Reviewer 1 ·

Basic reporting

no comment

Experimental design

no comment

Validity of the findings

no comment

Additional comments

no comment